# Spatial and temporal distribution characteristics and influencing factors of cultural heritage: A case of the Grand Canal (East Zhejiang section)-maritime silk road

Jie Li[1], Xinlian Yang[1], Yuhe Gao[2], Chao Gao[1,3]*

**1** Department of Geography and Spatial Information Techniques, Ningbo University, Ningbo, Zhejiang, China, **2** Department of Biomedical Informatics, University of Pittsburgh, Pittsburgh, Pennsylvania, United States of America, **3** Donghai Academy, Ningbo University, Ningbo, Zhejiang, China

* gaoqinchao1@163.com

## Abstract

The Grand Canal and the Maritime Silk Road in China are globally significant cultural routes, which have contributed a wealth of cultural heritage through their historical development. The study on the cultural heritage of the Grand Canal (East Zhejiang section)-Maritime Silk Road is of great significance for constructing the Grand Canal Cultural Belt and advancing the Belt and Road Initiative. Focusing on the Ningbo area, this study analyzes the spatial and temporal distribution of 1,755 cultural heritage sites over five historical periods and explores the influencing factors through spatial and statistical analysis. The results show that: (1) Ancient buildings, along with modern important historical sites and representative buildings, are the most numerous. The total number of cultural heritages shows an upward trend before the modern period, peaking in the Ming to Qing period. (2) The cultural heritage exhibits an overall aggregated spatial distribution, with varying patterns across different types. The Three-River Estuary is the high-density core area, with the number and density of cultural heritage decreasing as its distance increases. (3) Distribution characteristics of cultural heritage vary across different periods. More recent cultural heritage is increasingly concentrated around the Three-River Estuary. Over time, the center of gravity of cultural heritage has shifted sequentially to the south, southeast, west, and north. (4) The cultural heritage tends to be distributed in plain with low altitude and small slope, and shows strong hydrophilicity.

## Introduction

The concept of cultural heritage was first introduced by the United Nations Educational, Scientific and Cultural Organization (UNESCO) in 1972, encompassing both tangible and intangible cultural heritage [1]. Tangible cultural heritage is defined as cultural relics of historical, artistic, and scientific value, including immovable and movable cultural relics [2]. It represents the collective wisdom and creativity of humankind and serves as an invaluable global resource [3]. The distribution of cultural heritage worldwide reflects the complex pathways of human

**Data availability statement:** All relevant data are within the manuscript and its Supporting Information files.

**Funding:** This research is supported by Ningbo Key Research and Development Program of Ningbo Municipal Bureau of Science and Technology, Grant No. 2023Z137.

**Competing interests:** The authors have declared that no competing interests exist.

social development, geographic changes, and historical evolution, showcasing diverse temporal and spatial characteristics. The People's Republic of China General Secretary, Xi Jinping, has emphasized that cultural relics and heritage carry the genes and bloodline of the Chinese nation, representing non-renewable and irreplaceable resources of China's outstanding civilization. Research into the spatial and temporal distribution characteristics and influencing factors of cultural heritage not only aids in the understanding of the origin and development of human civilization, but also provides valuable insights for cultural preservation, tourism development and sustainable socio-economic growth.

The Grand Canal and the Maritime Silk Road have played significant roles in transportation, economic trade, and cultural exchanges throughout Chinese history. They serve as evidence of the far-reaching origins and long-lasting history of Chinese civilization. The Grand Canal is the earliest, largest, and longest man-made canal in the world, with a history that can be traced back to 486 B.C.E.. It consists of three major sections: the Sui-Tang Grand Canal, the Beijing-Hangzhou Grand Canal, and the East Zhejiang Grand Canal. Its total length is approximately 3,200 kilometers, extending from the north (Beijing) to the south (Zhejiang). The Grand Canal was a pivotal link between the northern and southern regions of China, marking the first historical instance when these regions were connected [4]. Furthermore, it acted as a crucial conduit between the Land Silk Road and the Maritime Silk Road [5]. The Maritime Silk Road, refers to the ancient East-West maritime trade and transportation routes. The Maritime Silk Road within China consisted of three main ports: Guangzhou, Quanzhou and Ningbo. The Maritime Silk Road connected China to more than a hundred countries in Asia, Africa and Europe, establishing significant links between China and the rest of the world [6]. The Grand Canal and the Maritime Silk Road facilitated the exchange and blending of various countries and regions in terms of production, lifestyles, customs, ideologies, social concepts, arts, aesthetics, religious beliefs, and philosophical thoughts [7,8]. This, in turn, promoted the overall development of politics, economy, science, technology, and culture in the regions along these routes [9–12].

In recent years, cultural heritage has become a popular area of academic interest. Scholars have conducted numerous studies on the spatial and temporal distribution and influencing factors of cultural heritage by employing geographic analysis methods at various scales. These studies have primarily focused on the single-provincial scale [13], inter-provincial scale [14,15], watershed scale [16], national scale [17,18], and even the global scale [19–21]. Current research on the cultural heritage of the Grand Canal and the Maritime Silk Road primarily focuses on the historical and social changes of the heritage [22], the religious, folklore, literary, and artistic cultural heritage [1,23], the types, values and protection of cultural heritage [24–27], and the construction of cultural heritage corridors [28,29]. Nevertheless, there is a paucity of research on the spatial and temporal evolution of the cultural heritage associated with the Grand Canal and the Maritime Silk Road. This study will help clarify the development of these two cultural routes and foster cultural preservation.

In light of these considerations, this study has selected Ningbo, a city situated at the confluence of the Grand Canal and the Maritime Silk Road, as the research area. The tangible cultural heritage of Ningbo has been identified as the study object, and an empirical study has been conducted utilizing nearest neighbor index (NNI), kernel density estimation (KDE), standard deviation ellipse (SDE), and buffer analysis. This study explores and summarizes the spatial and temporal distribution characteristics of the cultural heritage of the Grand Canal (East Zhejiang section)-Maritime Silk Road and its influencing factors from a geographical perspective. The findings can provide a scientific basis and technical support for cultural heritage development planning, and a useful reference for the high-quality sustainable development of the Grand Canal and the Maritime Silk Road.

## Materials and methods

### Research area

Located on the Ningshao Plain in the eastern part of Zhejiang Province (Fig 1), Ningbo, as the southernmost city of the Grand Canal and a key starting port of the Maritime Silk Road, holds a unique historical position. At the opening ceremony of the Belt and Road Forum for International Cooperation, General Secretary Xi Jinping stated that the ancient ports in Ningbo and other places are "living fossils" that record the history of the ancient Silk Road. Ningbo has a long history and is the birthplace of the Hemudu Culture, which represents over 7,000 years of civilization [30]. Due to its unique historical background and geographic location, Ningbo has developed a series of cultures with regional characteristics, mainly the Grand Canal culture and the Maritime Silk Road culture. As a carrier of cultural dissemination and historical memory, the cultural heritage in Ningbo bears the imprints of different

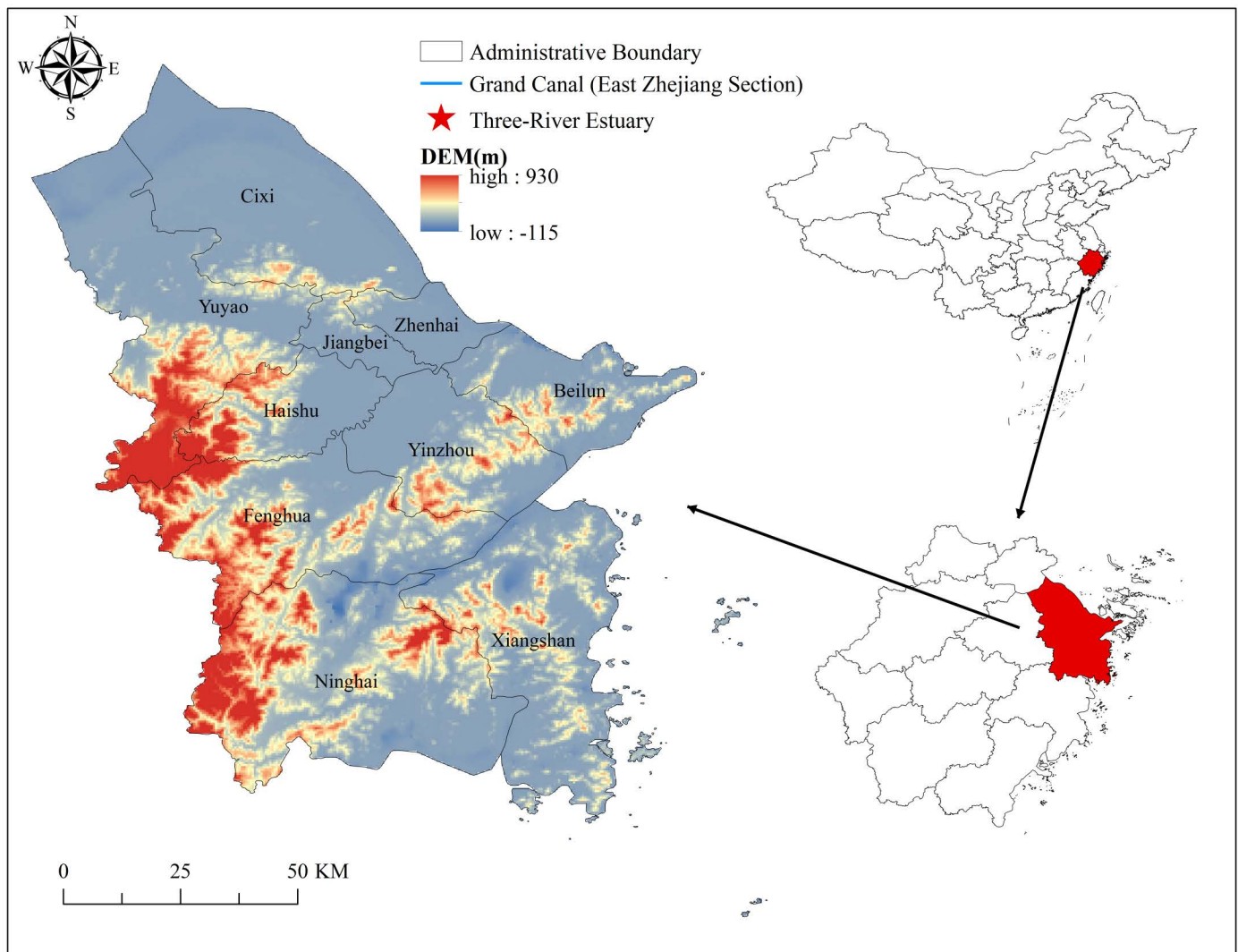

**Fig 1. Location and topographic features of the research area.** Note: The map is based on the standard map with review number GS (2020) 4619 downloaded from the website of Standard Map Service of the Ministry of Natural Resources of China (http://bzdt.ch.mnr.gov.cn/), produced by ArcGIS 10.8 software, with no modification of the base map.

eras, making it quintessential area for researching the cultural heritage of the Grand Canal and the Maritime Silk Road. As a carrier of cultural transmission and historical memory, cultural heritage carries the imprints of different eras. Therefore, Ningbo is selected in this study as a case to explore the cultural heritage of the Grand Canal (East Zhejiang section)-Maritime Silk Road.

## Data source

The data used in this study was obtained from the National Cultural Heritage Administration (http://www.ncha.gov.cn/) and the Zhejiang Cultural Heritage Administration (http://wwj.zj.gov.cn/). The dataset includes information on 33 state-protected historic sites, 103 province-protected historic sites, 497 municipality-protected historic sites, and 1,094 protected historic sites. Some cultural heritage items are listed at both the national and provincial levels, and some single items include multiple geographically distant sites. To more accurately study the spatial distribution characteristics of the cultural heritage, this study treats duplicated items as a single item, and single items containing multiple cultural heritages as multiple items. After merging and splitting, a total of 1,755 cultural heritage items were identified into six types: ancient buildings, ancient sites, ancient tombs, grotto temples and stone carvings, modern important historical sites and representative buildings, and others. Based on ancient Chinese historical stage, characteristics of local historical development, and existing research [15,17], the periods were divided as following: prehistory to pre-Qin period(Before 221 B.C.E.), Qin to Tang period(221 B.C.E.-960 C.E.), Song to Yuan period(960 C.E.-1368 C.E.), Ming to Qing period(1368 C.E.-1911 C.E.), and modern period(After 1911 C.E.).

Data attributes of cultural heritage include name, era, type, latitude and longitude coordinates, address, protection level, and other relevant information of the cultural heritage. The geographic coordinates of cultural heritage sites were primarily collected using Global Positioning System (GPS) during field trips. Some hard-to-reach cultural heritage sites had their coordinates obtained by applying Baidu map coordinate picking. Additionally, Google Earth was utilized to cross-check these coordinates. Excel and ArcGIS10.8 software were used to establish the database of the cultural heritage of the Grand Canal (East Zhejiang section)-Maritime Silk Road and create the spatial distribution map of the cultural heritage (Fig 2). The Digital Elevation Model (DEM) data and the river data were obtained from the Geospatial Data Cloud (http://www.gscloud.cn/) and the Open Street Map (https://www.openstreetmap.org/), respectively. In addition, for the river data, irrelevant factors such as drains and sewers were eliminated and naturally occurring rivers were selected.

## Research method

**NNI.** Cultural heritage can be represented as point elements on a map, and their distribution in geospatial space can be categorized into three types: agglomerative, uniform, and random. The NNI can be used to reflect the spatial distribution pattern of cultural heritage. This index is the ratio of the actual near neighbor distance to the expected near neighbor distance [31]. The formula for calculating the NNI is as follows:

$$NNI = \frac{\overline{r_1}}{r_0} = \frac{\left\{\sum_{i=1}^{n} d_i\right\}/n}{0.5/\sqrt{n/A}} \tag{1}$$

where $\overline{r_1}$ represents the average value of the nearest-neighbor actual distance of cultural heritage sites; $r_0$ represents the expected value of the nearest-neighbor distance; $d_i$ is the

# a.different types     b.different periods

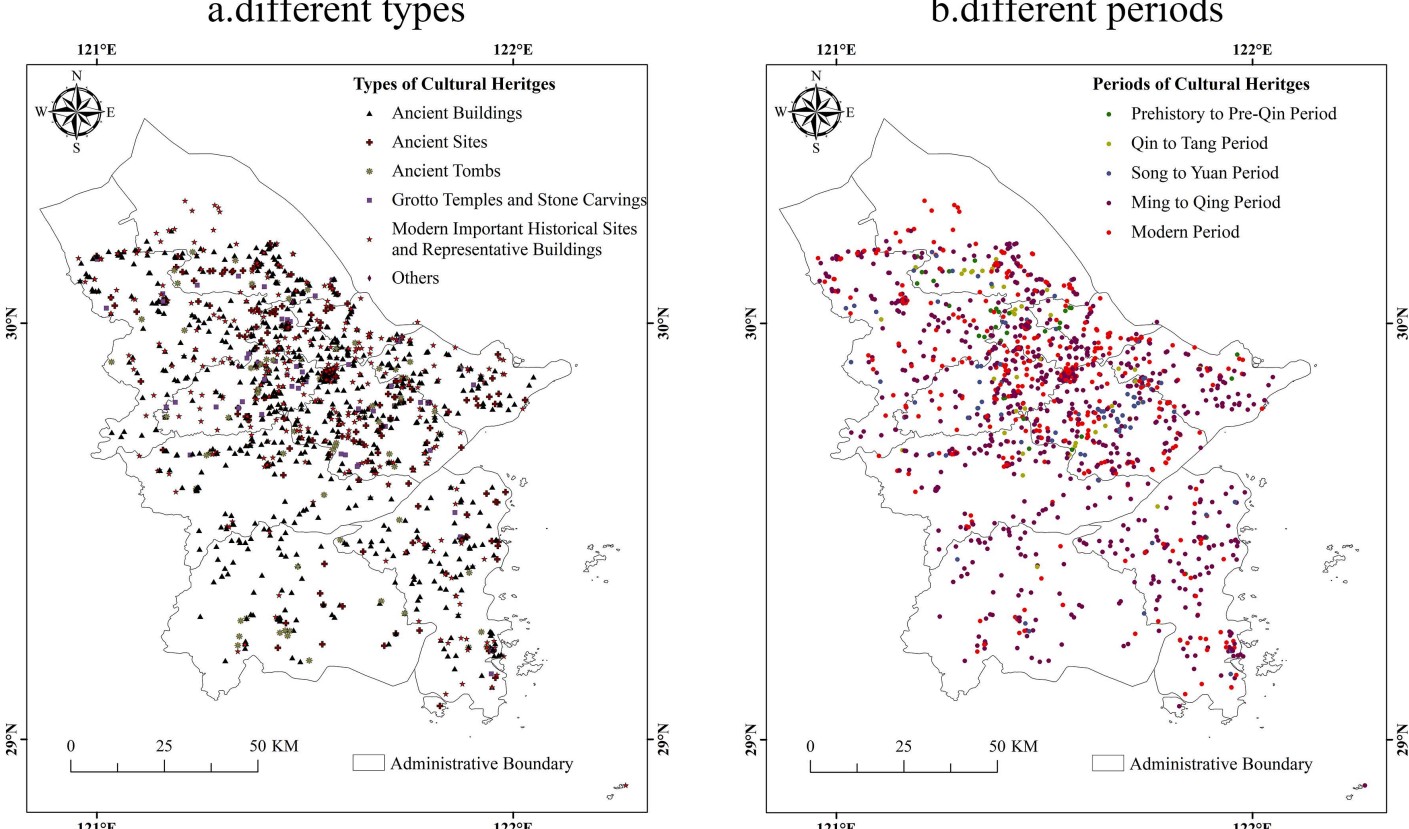

**Fig 2. Spatial distribution of the grand canal (East Zhejiang section)-maritime silk road cultural heritage.** Note: The map is based on the standard map with review number GS (2020) 4619 downloaded from the website of Standard Map Service of the Ministry of Natural Resources of China (http://bzdt.ch.mnr.gov.cn/), produced by ArcGIS 10.8 software, with no modification of the base map.

nearest-neighbor actual distance; $n$ is the number of cultural heritage sites; and $A$ is the area of Ningbo City [32].

**KDE.** KDE can clearly represent the spatial distribution and aggregation characteristics of geographical elements [15]. The method uses each point element as the center of a circle and the radius as the density value of the unit circle. It analyzes the continuous change of spatial density within the unit area. The kernel density is used to represent the density of spatial distribution and local characteristics of cultural heritage, and to reflect the degree of spatial aggregation of cultural heritage. The formula for calculating the kernel density is as follows:

$$f(x) = \frac{1}{nh} \sum_{i=1}^{n} k(\frac{x - x_i}{h}) \qquad (2)$$

where $k\left(\frac{x - x_i}{h}\right)$ is the kernel function; $h(>0)$ is the search radius (or bandwidth); $n$ is the number of known points within the bandwidth; and $(x - x_i)$ is the distance from the estimated point $x$ to the sample point $x_i$. The distribution density of the point elements increases as the value of $f(x)$ increases, and decreases as the value of $f(x)$ decreases [33].

**SDE.** The SDE captures the central tendency, dispersion, and directional patterns of spatial elements, enabling the analysis of cultural heritage distribution over different periods [34]. The formula for calculating the SDE is as follows:

$$SDE_x = \sqrt{\frac{\sum_{i=1}^{n}(x_i - \overline{X})^2}{n}}, SDE_y = \sqrt{\frac{\sum_{i=1}^{n}(y_i - \overline{Y})^2}{n}} \qquad (3)$$

where $x_i$ and $y_i$ represent the longitude and latitude of the cultural heritage $i$, respectively, and $n$ represents the number of cultural heritages in a particular period.

## Buffer analysis

Buffer analysis is a spatial analysis method used to explore the impact of a geographic entity on its surrounding features and to determine the adjacency between different geographic elements [35]. This study takes the natural river as the object of analysis and the cultural heritage sites as the affected objects, establishes a buffer within a certain distance around the river, and calculates the number of cultural heritage sites in the buffer through overlay analysis. The calculation formula for the buffer analysis is as follows:

$$D = \left\{ X \middle| d(x,S) \leq R \right\} \qquad (4)$$

where $x$ is the affected object of the buffer analysis; $S$ is the given object for establishing the buffer; $d$ represents the Euclidean distance between $x$ and $S$; and $R$ is the radius of the buffer.

## Results

### Type structure and distribution characteristics of cultural heritage

**Composition structure and volume ratio.** In terms of types (Table 1), ancient buildings and modern important historical sites and representative buildings are the first two major cultural heritage, totaling 957 (54.53%) and 490 (27.92%) respectively, followed by ancient sites (143, 8.15%), ancient tombs (84, 4.79%) and grotto temples and stone carvings (74, 4.22%), and others only presents 7 (0.40%) cases. The cultural heritage of the Grand Canal (East Zhejiang section)-Maritime Silk Road shows the structural characteristics of ancient buildings and modern important historical sites and representative buildings, supplemented by ancient sites, ancient tombs, and grotto temples and stone carvings, also including few other rare types, such as caves and terraces.

In terms of total number divided by periods (Table 1, Fig 3), Ming to Qing period has the highest number of cultural heritages, up to 1076 (61.31%), followed by modern period cultural heritages with 478 (27.24%), and less cultural heritages in Song to Yuan (110, 6.27%), Qin to

**Table 1. Numbers-periods-types of the Grand Canal (East Zhejiang section)-Maritime Silk Road cultural heritage.**

| Types | Ancient buildings | Ancient sites | Ancient tombs | Grotto temples and stone carvings | Modern important historical sites and representative buildings | Others | Total |
|---|---|---|---|---|---|---|---|
| Prehistory to pre-Qin period | 1 | 34 | 5 | 0 | 0 | 0 | 40 |
| Qin to Tang period | 8 | 28 | 14 | 1 | 0 | 0 | 51 |
| Song to Yuan period | 39 | 26 | 17 | 27 | 0 | 1 | 110 |
| Ming to Qing period | 886 | 55 | 43 | 34 | 57 | 1 | 1076 |
| Modern period | 23 | 0 | 5 | 12 | 433 | 5 | 478 |
| Total | 957 | 143 | 84 | 74 | 490 | 7 | 1755 |

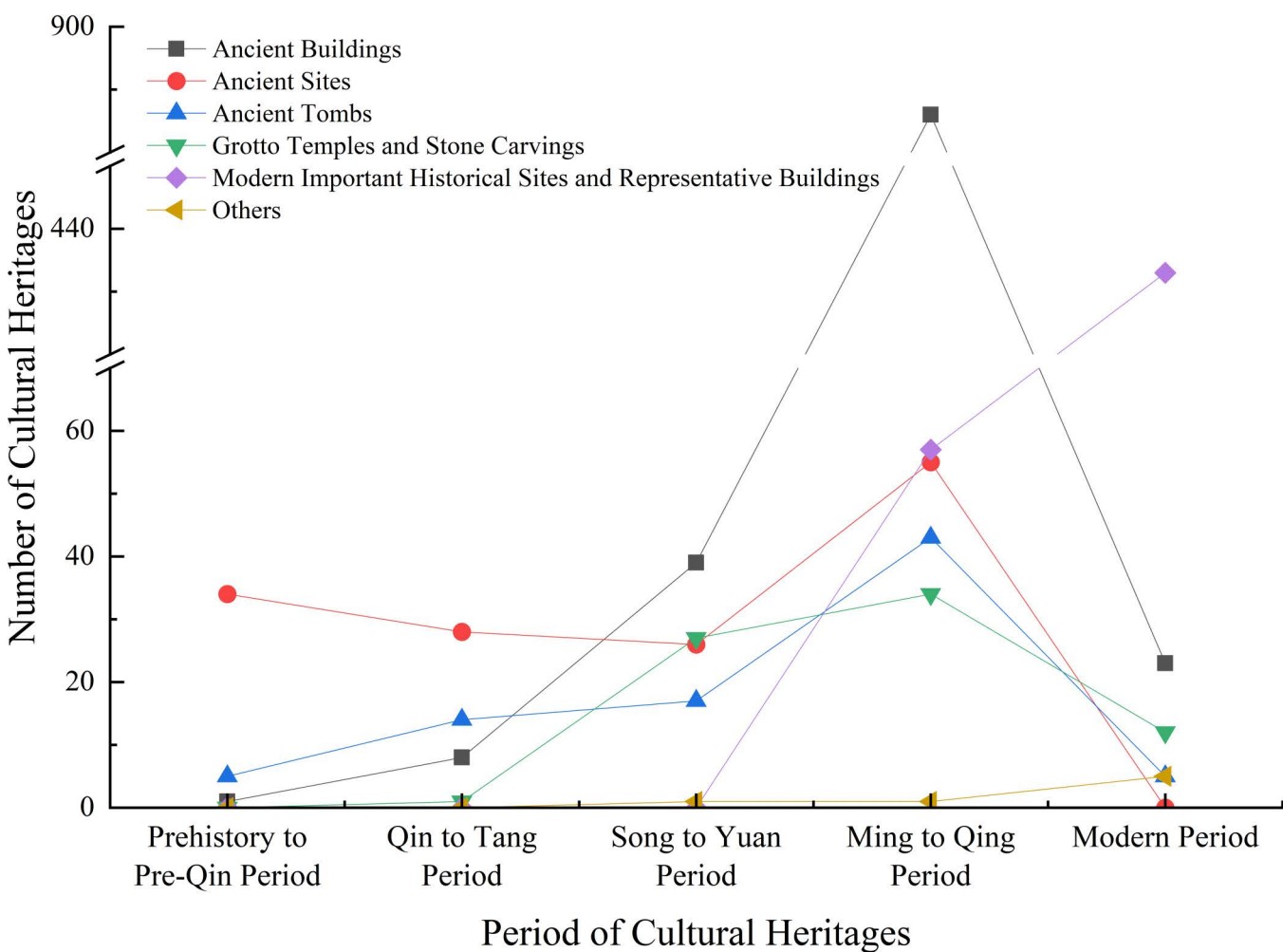

**Fig 3. Changes in the number of cultural heritages over time.**

Tang (51, 2.91%), and prehistory to pre-Qin periods (40, 2.28%). In general, the number of current existing cultural heritage before modern period shows an overall upward trend and reaches its peak in Ming to Qing period. Cultural heritage is a non-renewable resource. The age-old cultural heritage is mainly made of earth and wood structures, which is vulnerable to both natural erosion and human destruction. However, as time progresses, advancements in construction techniques and protection technologies, along with improved socio-economic conditions, have resulted in better preservation and increased numbers of cultural heritage remains.

**Spatial distribution characteristics.**

(1) Spatial distribution patterns

The Average Nearest Neighbor tool in ArcGIS 10.8 software was used to calculate the NNI of the cultural heritage of the Grand Canal (East Zhejiang section)-Maritime Silk Road (Table 2). It is generally believed that NNI ≤ 0.5 indicates an aggregated distribution, NNI ≥ 1.5 indicates a uniform distribution, 0.5 < NNI ≤ 0.8 indicates an aggregated-random distribution, 0.8 < NNI < 1.2 indicates a random distribution, and 1.2 ≤ NNI < 1.5 indicates a random-discrete distribution [36]. As shown in Table 2, the overall NNI of the cultural heritage of the Grand

**Table 2. NNI of the Grand Canal (East Zhejiang section)-Maritime Silk Road cultural heritage.**

| Types | Number | Z-score | P-value | NNI | Distribution type |
|---|---|---|---|---|---|
| Ancient buildings | 957 | −28.003 | 0 | 0.527 | Aggregated-random distribution |
| Ancient sites | 143 | −7.274 | 0 | 0.682 | Aggregated-random distribution |
| Ancient tombs | 84 | −5.102 | 0 | 0.709 | Aggregated-random distribution |
| Grotto temples and stone carvings | 74 | −6.993 | 0 | 0.575 | Aggregated-random distribution |
| Modern important historical sites and representative buildings | 490 | −22.205 | 0 | 0.476 | Aggregated distribution |
| Others | 7 | −0.907 | 0.365 | 0.821 | Random distribution |
| Total | 1755 | −44.793 | 0 | 0.441 | Aggregated distribution |

Canal (East Zhejiang section)-Maritime Silk Road is 0.441, which indicates that a strong aggregated distribution pattern. However, different types of cultural heritage exhibit varying distribution patterns. The modern important historical sites and representative buildings have the strongest aggregation, with the NNI of 0.476. Ancient buildings, ancient sites, ancient tombs, grotto temples and stone carvings all show an aggregated-random distribution pattern, while ancient tombs present the weakest aggregation, with the NNI of 0.709. The NNI of others is higher than 0.8, which indicates that there is no significant difference between their overall distribution and random distribution, while the sample size may be too small (7) to accurately reflect the overall distribution.

(2) Spatial aggregation characteristics

The Kernel Density tool (with a bandwidth of 10km) in ArcGIS 10.8 software was used to calculate the kernel density of the cultural heritage of the Grand Canal (East Zhejiang section)-Maritime Silk Road. This analysis helps to derive the overall spatial aggregation status and the characteristics of different types, clarifying the difference and complementarity of cultural heritage resources. As a whole (Fig 4), the cultural heritage formed a high-density core area and two sub-core areas. Among them, the high-density core area is located at the junction of Haishu, Jiangbei and Yinzhou, known as the Three-River Estuary. The two sub-core areas, influenced by this core area, include the center of urban area in Yuyao and the fishing port area of Shipu in Xiangshan. The spatial distribution of cultural heritage is closely related to the natural and human factors. The Three-River Estuary is not only the most densely populated and economically prosperous zone in modern Ningbo, but also the center of the old city of Ningbo which has been continuing since the Tang Dynasty [37]. Its advantages of low topography, convenient transportation, prosperous trade and culture have made the Three-River Estuary become the center of gravity for the cultural heritage of the Grand Canal (East Zhejiang section)-Maritime Silk Road.

In terms of different types of cultural heritage (Fig 5), the aggregation characteristic of ancient buildings is similar to the overall cultural heritage, forming a core area in the Three-River Estuary and radiating out to the surrounding area, the specific forms of which include residences, ancestral halls, ancient bridges and temples. Ancient sites formed three high-density core areas in Yuyao Jiangbei junction, southern Haishu and Dongqian Lake area of central Yinzhou, meanwhile, sub-core areas were formed in Shanglin Lake area of southern Cixi and southeastern Beilun. Ancient tombs were more widely distributed, with a core area in northern Haishu. Grotto temples and stone carvings and modern important historical sites and representative buildings, are both concentrated in north-central region of Ningbo, showing uneven distribution. The former centers around the Three-River Estuary, Haishu and Yinzhou Beilun junction, while the latter's distribution tends to align with overall pattern. The cultural heritage of others is scarce in number, resulting in a large difference in distribution characteristics.

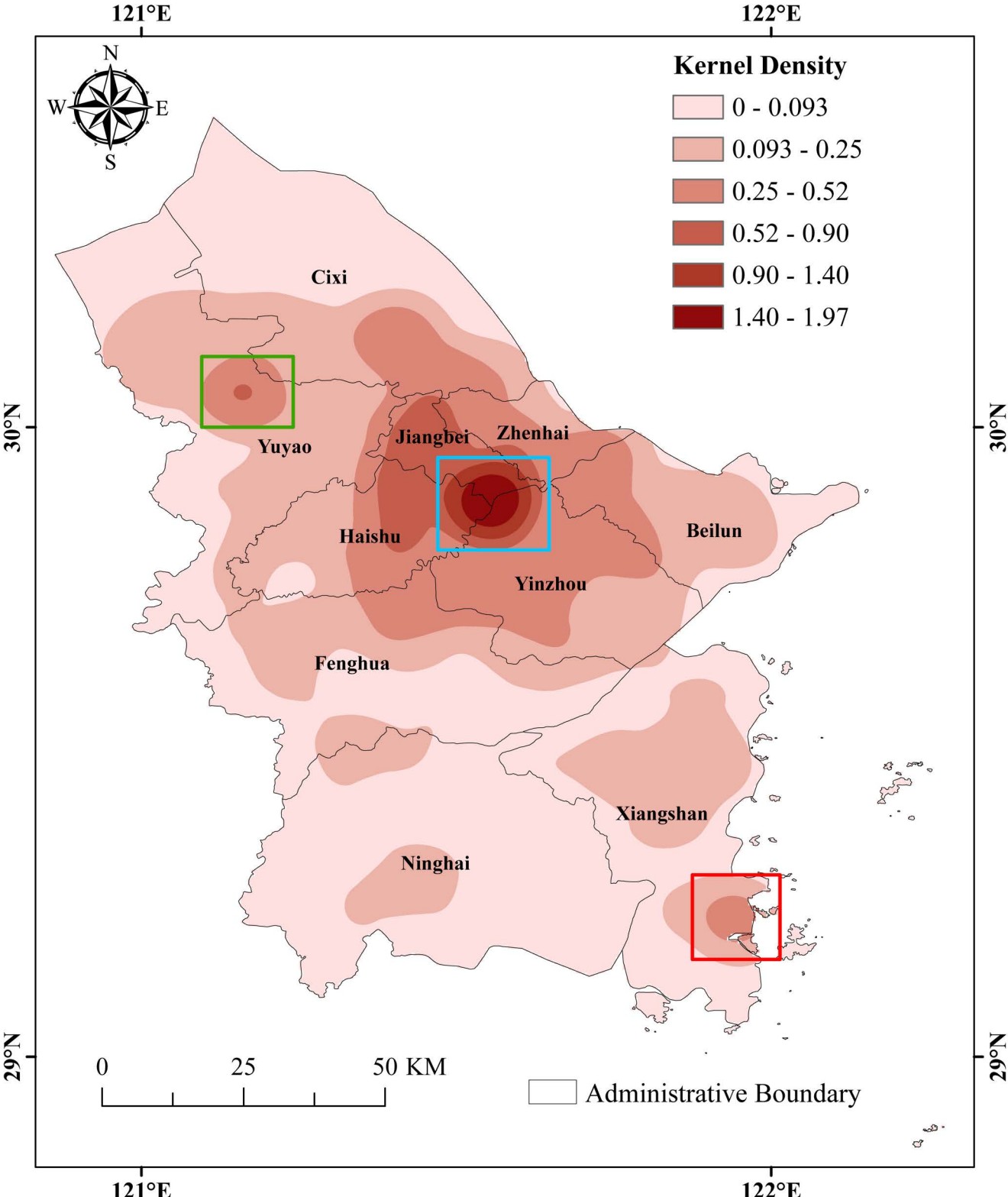

**Fig 4. Overall kernel density of the Grand Canal (East Zhejiang section)-Maritime Silk Road cultural heritage.** (The blue box indicates Three-River Estuary, the green box indicates Yuyao city center, and the red box indicates Xiangshan Shipu fishing port.) Note: The map is based on the standard map with review number GS (2020) 4619 downloaded from the website of Standard Map Service of the Ministry of Natural Resources of China (http://bzdt.ch.mnr.gov.cn/), produced by ArcGIS 10.8 software, with no modification of the base map.

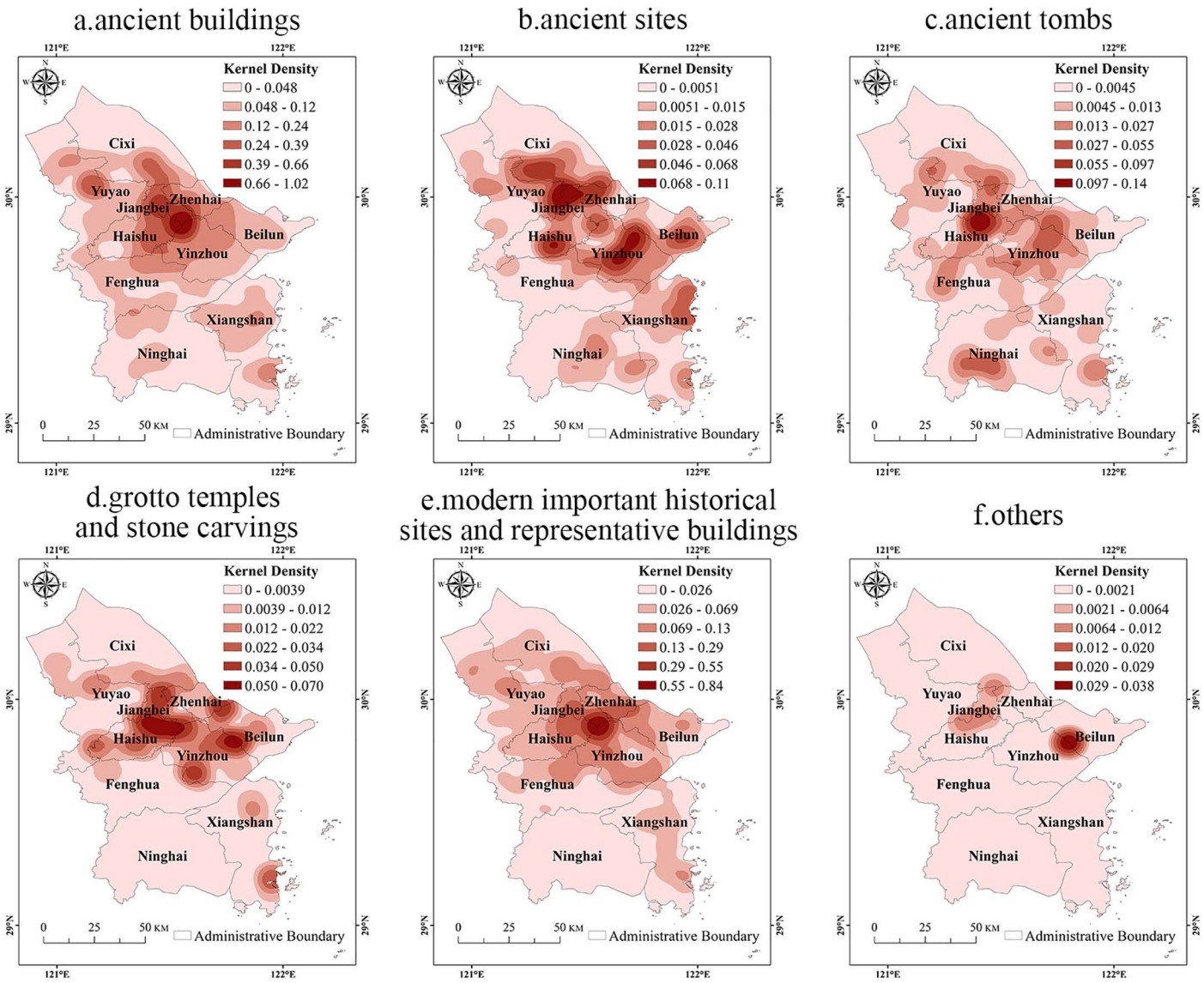

**Fig 5. Kernel density of different types of the Grand Canal (East Zhejiang section)-Maritime Silk Road cultural heritage.** Note: The map is based on the standard map with review number GS (2020) 4619 downloaded from the website of Standard Map Service of the Ministry of Natural Resources of China (http://bzdt.ch.mnr.gov.cn/), produced by ArcGIS 10.8 software, with no modification of the base map.

The above features indicate both difference and similarity in the spatial distribution of different types of cultural heritage resources, the influence of geographical law on their distribution [38]. In addition, the internal characteristics and external requirements of different types of cultural heritage resources also have a certain influence on the spatial distribution pattern of cultural heritage, which is also one of the important causes for the characteristics of the cultural heritage spatial aggregation [36].

**Temporal distribution characteristics.** To reveal the spatial and temporal evolution characteristics of the cultural heritage of the Grand Canal (East Zhejiang section)-Maritime Silk Road, we conducted kernel density analyses and statistical collations for each of the five periods (Table 3, Fig 6). The spatial distribution of cultural heritage is concentrated in north-central region of Ningbo. The closer the period is to the present, the more cultural heritage

**Table 3. Distribution and composition of the Grand Canal (East Zhejiang section)-Maritime Silk Road cultural heritage in different periods.**

| Periods | Main distribution areas | Main types | Representative cultural heritages |
| --- | --- | --- | --- |
| Prehistory to pre-Qin period | Yuyao Hemudu area | Ancient sites | Hemudu Site, Jintoushan Site |
| Qin to Tang period | Cixi Shanglin Lake area | Ancient sites, ancient tombs | Shanglin Lake Yue Kiln Sites, Tiantong Temple, Tuoshan Weir |
| Song to Yuan period | Yinzhou Dongqian Lake area | Ancient buildings, ancient sites, grotto temples and stone carvings | Yongfengku Site, Dongqian Lake Tomb Complex |
| Ming to Qing period | Three-River Estuary | Ancient buildings | Tianyi Pavilion, Qing'an Guild Hall, Zhenhaikou Coast Defense Site |
| Modern period | Three-River Estuary | Modern important historical sites and representative buildings | The Former Site of East Zhejiang Anti-Japanese Base, Qianye Guild Hall |

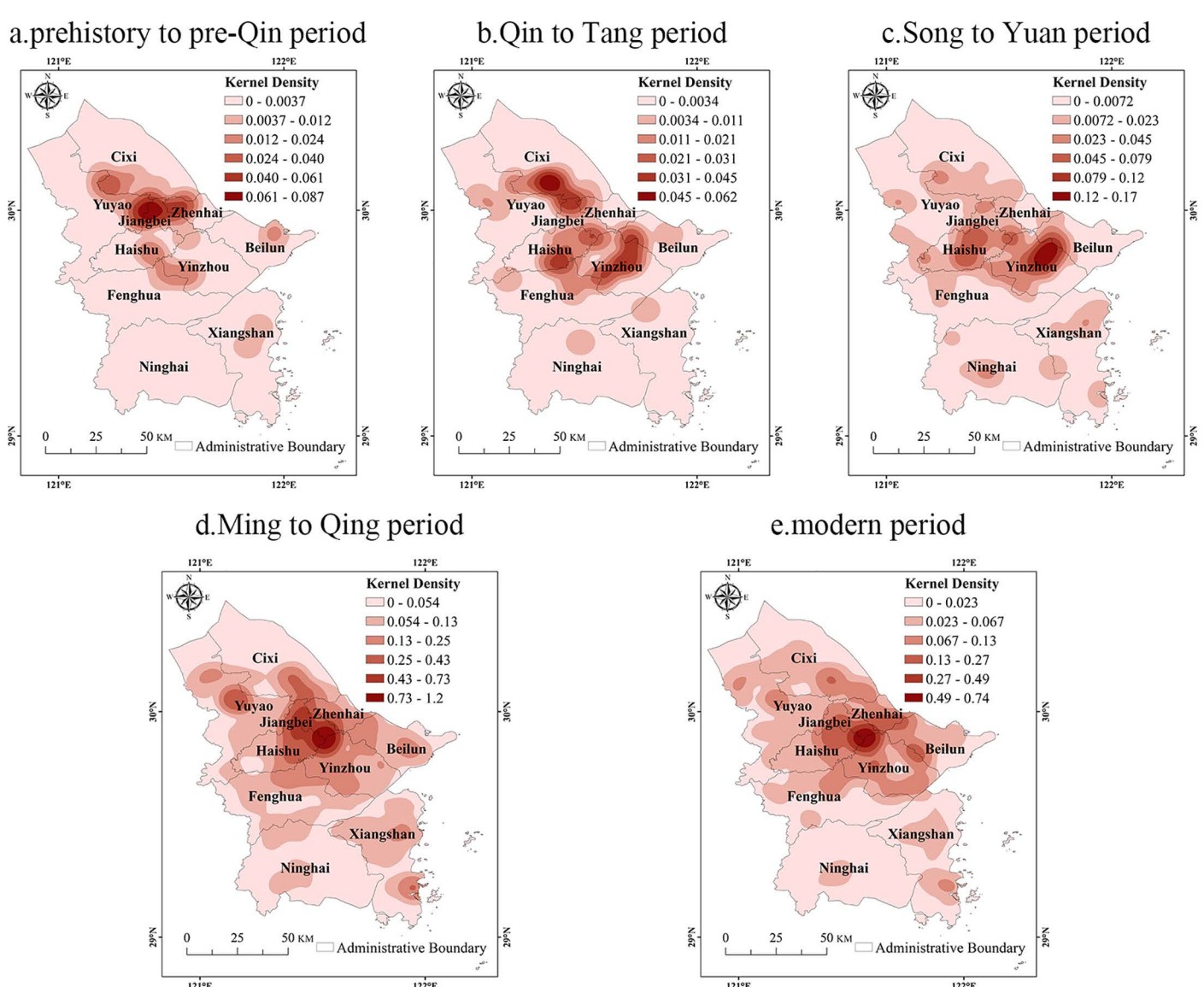

**Fig 6. Kernel density of the Grand Canal (East Zhejiang section)-Maritime Silk Road cultural heritage in different periods.** Note: The map is based on the standard map with review number GS (2020) 4619 downloaded from the website of Standard Map Service of the Ministry of Natural Resources of China (http://bzdt.ch.mnr.gov.cn/), produced by ArcGIS 10.8 software, with no modification of the base map.

gathers in the Three-River Estuary, which was the most densely distributed spatial area of cultural heritage since Ming to Qing period.

(1) Prehistory to pre-Qin period

During prehistory to pre-Qin period, the majority of cultural heritage (85%) consisted of ancient sites, with 34 in total. These sites were mostly human settlements from the Neolithic Age, concentrated in Yuyao, Cixi, and Jiangbei. The core area of these settlements was located at Yuyao Jiangbei junction. The Jingtoushan Site is the earliest and deepest marine culture site found in China's coastal area [39]. It provides evidence of ancient coastal changes and reveals the cultural gene of Ningbo as a port city of the Maritime Silk Road. Additionally, the construction of the Grand Canal began during the Spring and Autumn Period (770 B.C.E.–476 B.C.E.), playing a significant role in Ningbo's early social development and cultural dissemination. Many sites and relics can be found along the Grand Canal [40].

(2) Qin to Tang period

During Qin to Tang period, the cultural heritage was primarily centered around Shanglin Lake area of southern Cixi. Two sub-core areas were also formed in Haishu and Dongqian Lake area of central Yinzhou. This period is characterized by numerous ancient sites, and an increase in ancient buildings and ancient tombs. Additionally, grotto temples and stone carvings began to emerge. During this period, 23 kiln sites were discovered, accounting for 45.10% of the total cultural heritage. These kilns produced mainly Yue celadon, which was transported from Ningbo port through the Grand Canal and then traded overseas via the Maritime Silk Road, which promoted the development of canal-sea linkage [41].

(3) Song to Yuan period

During Song to Yuan period, Dongqian Lake area of central Yinzhou was the high-density core area of cultural heritage, with Haishu and the Three-River Estuary as sub-core areas. The number of ancient buildings and grotto temples and stone carvings increased significantly during this period, while the number of ancient sites and ancient tombs remained relatively stable. Kiln sites, stone carvings, tomb groups, pagodas, ancient bridges and ancient roads were the main components of cultural heritage in this period. Among them, a large number of stone carvings, pagodas and tomb passages remain in Dongqian Lake area, which is the largest, most numerous and best-carved Southern Song tomb passage stone carvings remaining in China [42]. In Yuan Dynasty, Ningbo implemented an opening-up policy. Yongfengku Site was set up at that time to specialize in managing overseas trade and domestic trade taxes and confiscation [43]. Its ruins and unearthed artifacts have become an important basis for proving that the prosperity of the Maritime Silk Road in Ningbo at that time.

(4) Ming to Qing period

During Ming to Qing period, there were a great number of ancient buildings (886), accounting for 82.34% of the cultural heritage of this period. The cultural heritage of this period was concentrated in the Three-River Estuary, with clusters of ancient buildings all along the coast. Meanwhile, the forms of ancient buildings became various, including palace, temple, bridge, pavilion, building, residence, and mansion; the functions also became comprehensive, involving various aspects including life, production, military defense, water conservancy, transportation, external exchanges, and religious beliefs. The Qing'an Guild Hall is a representative building of this period, witnessing the connection between the Grand Canal and the Maritime Silk Road. The interior of the guild hall is divided into two sections, with the front theater dedicated to the worship of Mazu and the rear theater used for the performance of

plays during the gatherings of merchant gangs. In addition, under the influence of the war of aggression, a large number of coastal defense sites, consulates with colonial colors, European-style architectural complexes and Catholic churches appeared in Ming to Qing period [44], symbolizing the decline of the Maritime Silk Road in the latter period.

(5) Modern period

The Three-River Estuary remains the high-density core area of the cultural heritage in modern period. Overall, the spatial distribution pattern continues the characteristics of Ming to Qing period. This period is dominated by modern important historical sites and representative buildings (433), accounting for 90.59% of the total cultural heritage in this period. Influenced by the historical characteristics of this period, Ningbo, as one of the earliest trading ports in China and one of the bases of the anti-Japanese revolution, is characterized by representative heritages such as the former residences of celebrities, mansions, residential houses, red cultural heritage, monumental buildings and Western buildings.

**Center of gravity and evolutionary trends.** The Directional Distribution tool in ArcGIS 10.8 software was used to obtain the SDE of cultural heritage distribution (Fig 7). In general, the azimuths of the SDEs of cultural heritage in different periods were between 138° and 148.5° (Table 4), showing a northwest-southeast distribution pattern. In terms of the evolutionary trend, the center of gravity of cultural heritage in five periods experienced a south-southeast-west-north transfer trajectory, and continuously converged to the Three-River Estuary as a whole. The center of gravity during prehistory to pre-Qin period was located in Jiangbei. It then shifted southward to Haishu during Qin to Tang period, and further southward to Yinzhou during Song to Yuan period and Ming to Qing period. Finally, it moved northward to Haishu Yinzhou junction during modern period. The increasing length of the SDE's axes suggests a more pronounced directionality in the distribution of cultural heritage and a broader, more decentralized distribution range, which persisted into the modern period. The overall expanding distribution of cultural heritage benefits from the promotion of the Grand Canal and the Maritime Silk Road. The Grand Canal connected inland regions while the Maritime Silk Road linked overseas regions, which led to the prosperous development of navigation, economy and culture, and contributed to the continuous spread of cultural heritage in the canal regions and coastal areas.

## Influencing factors of cultural heritage distribution

The spatial and temporal distribution of the Grand Canal (East Zhejiang section)-Maritime Silk Road cultural heritage shows typical aggregation, with significant differences between different types and periods. Spatial differentiation results from a combination of factors such as nature and human environments. Drawing on existing studies [15,17,20,33,36], six indicators are selected to explore the influence mechanism of cultural heritage distribution through overlay analysis, buffer analysis, and literature analysis.

**Natural factors.**

(1) Topography and elevation

Ningbo is located on the third ladder of topography in China, with low elevations in the southwest and higher elevations in the northeast, sloping towards the East China Sea from southwest to northeast. The low mountainous and hilly area of east Zhejiang lies to the southwest, with the Siming Mountain Range, oriented southwest-northeast, traversing Yuyao, Fenghua and Yinzhou. To the south, a branch of Tiantai Mountain, which enters from the southwest of Ninghai and extends into the southern mountains through Xiangshan Harbor. The city's terrain is comprised of 24.9% mountain, 25.2% hill, 1.5% tableland, 8.1%

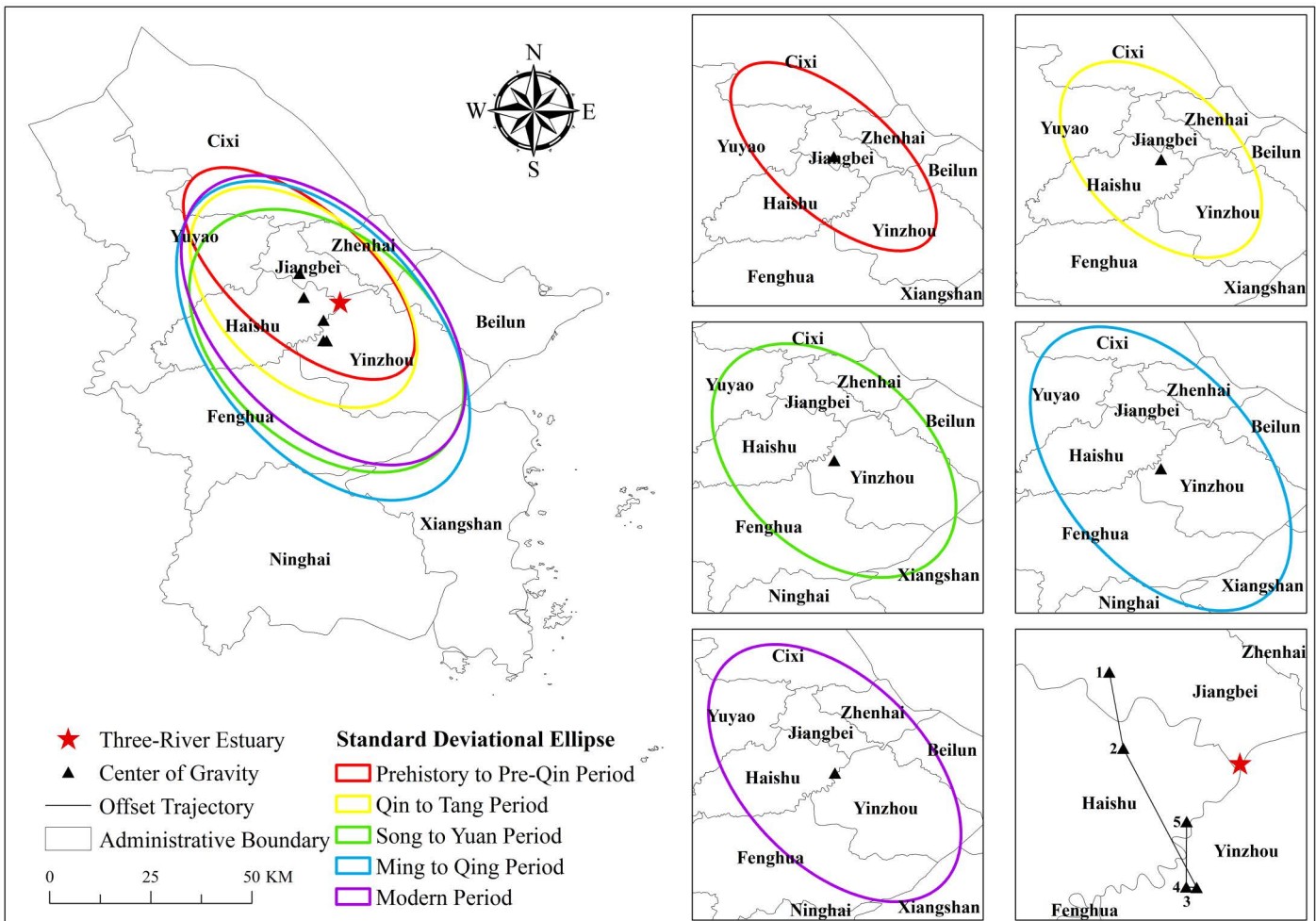

**Fig 7. Center of gravity and direction of the Grand Canal (East Zhejiang section)-Maritime Silk Road cultural heritage in different periods.** Note: The map is based on the standard map with review number GS (2020) 4619 downloaded from the website of Standard Map Service of the Ministry of Natural Resources of China (http://bzdt.ch.mnr.gov.cn/), produced by ArcGIS 10.8 software, with no modification of the base map.

**Table 4. Changes in SDE parameters of the Grand Canal (East Zhejiang section)-Maritime Silk Road cultural heritage in different periods.**

| Periods | Center of gravity coordinate | Long half shaft (km) | Short half shaft (km) | Azimuth (°) | Direction |
|---|---|---|---|---|---|
| Prehistory to pre-Qin period | 121.47°E, 29.94°N | 32.69 | 15.27 | 138.01 | Northwest-Southeast |
| Qin to Tang period | 121.48°E, 29.89°N | 31.31 | 18.83 | 142.56 | Northwest-Southeast |
| Song to Yuan period | 121.53°E, 29.79°N | 36.65 | 24.11 | 142.59 | Northwest-Southeast |
| Ming to Qing period | 121.52°E, 29.79°N | 43.80 | 25.06 | 148.50 | Northwest-Southeast |
| Modern period | 121.52°E, 29.84°N | 40.62 | 23.63 | 144.87 | Northwest-Southeast |

valley, and 40.3% plain. The elevation distribution of the cultural heritage was determined by overlaying their spatial information with the DEM (Fig 8a). The elevation statistics of the cultural heritage sites reveal a distinct distribution pattern (Table 5). The major number of cultural heritages in the elevation range of 0-5 meters, totaling 628 sites, and accounting for 35.78%, followed by the elevation ranges of 5-10 meters and 10-50 meters with 403 and 268 cultural heritages, accounting for 22.96% and 15.27%, respectively. The least number of

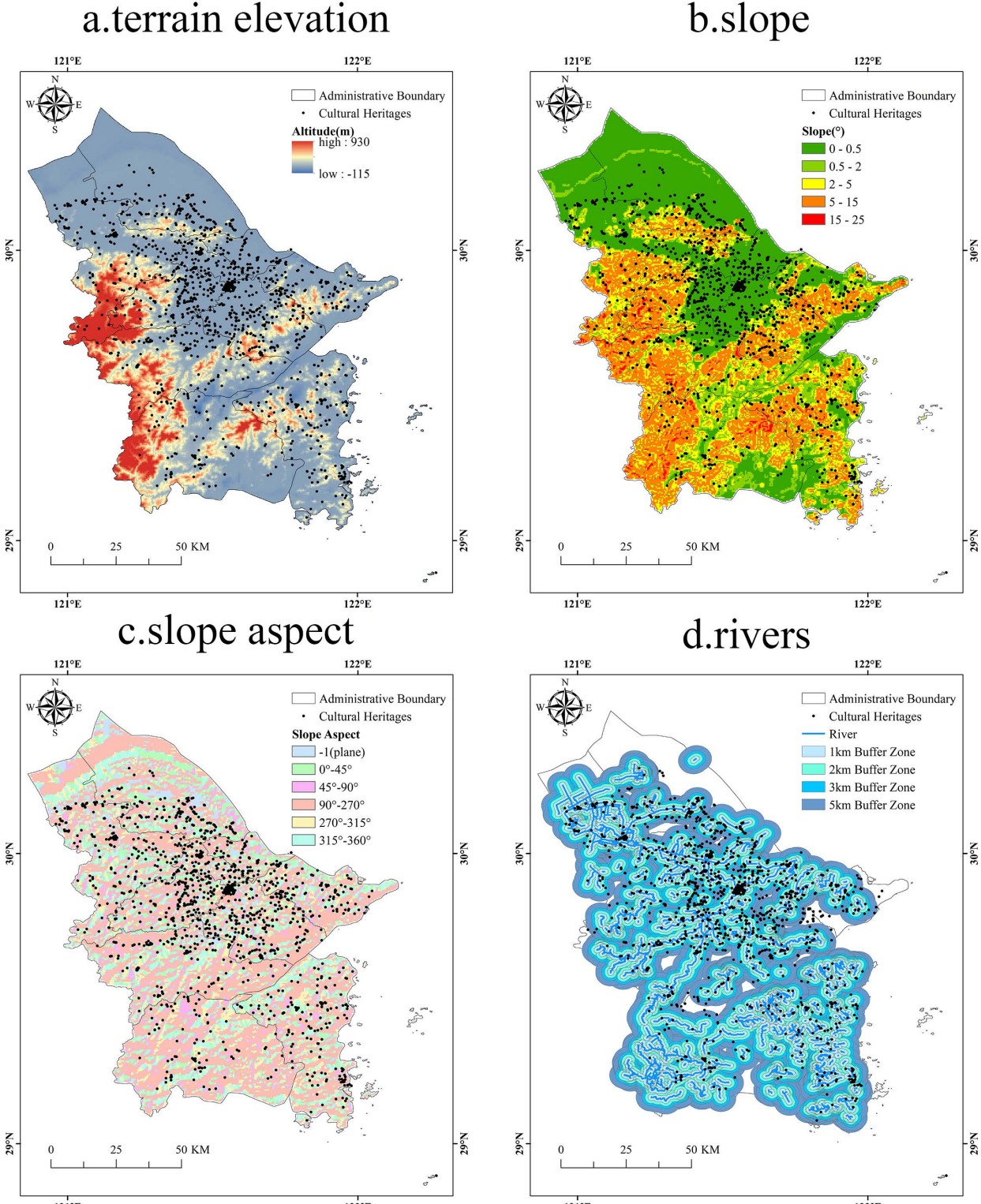

**Fig 8. Various natural factors and spatial distribution of the Grand Canal (East Zhejiang section)-Maritime Silk Road cultural heritage.**
Note: The map is based on the standard map with review number GS (2020) 4619 downloaded from the website of Standard Map Service of the Ministry of Natural Resources of China (http://bzdt.ch.mnr.gov.cn/), produced by ArcGIS 10.8 software, with no modification of the base map.

**Table 5. Number and proportion of the Grand Canal (East Zhejiang section)-Maritime Silk Road cultural heritage in different elevation ranges.**

| Altitude (m) | Number of cultural heritages | Proportion (%) |
|---|---|---|
| <0 | 15 | 0.85 |
| 0–5 | 628 | 35.78 |
| 5–10 | 403 | 22.96 |
| 10–50 | 268 | 15.27 |
| 50–100 | 151 | 8.60 |
| 100–200 | 142 | 8.09 |
| 200–500 | 126 | 7.18 |
| >500 | 22 | 1.25 |

cultural heritages in the ranges of less than 0 meters and more than 500 meters are 15 and 22, respectively.

Topography and elevation significantly affect the distribution of cultural heritage. Generally, the number of cultural heritage sites increases with elevation up to a certain point, after which it decreases. Plain areas are more suitable for human settlement, and as cities form and economies develop, cultural resources accumulate. In contrast, higher altitude areas pose challenges for transportation and human activities, and they may have various constraints that hinder the distribution of cultural heritage.

(2) Slope and slope aspect

The slope and slope aspect have a direct relationship with the terrain elevation distribution. The slope analysis map (Fig 8b) was generated using ArcGIS10.8 software based on the Ningbo elevation map. The slope aspect analysis map (Fig 8c) was further obtained, and the slopes and slope aspects of the cultural heritages were extracted (Table 6 and 7). According to the extraction statistics, 1,741 cultural heritages are distributed in plains, micro slopes, gentle slopes and slopes in the range of 0°-15° [45], accounting for 99.20% of the total. The mountainous topographic characteristics of southwestern Ningbo led to a certain number of cultural heritages being distributed in steep areas with slopes of 15° or more. In terms of slope aspect, according to existing study [46], the best light condition is at plane (−1); 90°–270° belongs to good light condition; 45°–90° and 270°–315° belong to poor light condition; and 315°–360° and 0°–45° belong to worst light condition. Overall, there are 922 cultural heritages in the best or good lighting conditions, accounting for 52.54%, while 399 and 434 cultural heritages are in the poor and worst lighting conditions, accounting for 22.74% and 24.73%, respectively.

These results indicate that the cultural heritage of the Grand Canal (East Zhejiang section)-Maritime Silk Road is mainly distributed in low-lying plains with small and gentle slopes. However, there is no clear relationship between the distribution of cultural heritage and slope aspect. During historical period, human activities were mainly concentrated in areas with gentle terrain, which facilitated the collection and utilization of natural resources. As a result, cultural heritage shows an obvious characteristic of gently sloping distribution. Although feng shui was considered important in the past China [47], where people believed in selecting addresses based on the principle of "carry the Yin and embrace the Yang, bound by mountains and near water" [48], the influence of slope aspect on the distribution of cultural heritage is not significant. It can be inferred that: the distribution of cultural heritage is more influenced by factors such as water sources and other natural resources [49]; specific cultural heritage, such as military defense buildings and water conservancy facilities, do not need to give priority to the impact of slope aspect.

**Table 6. Number and proportion of the Grand Canal (East Zhejiang section)-Maritime Silk Road cultural heritage in different slope ranges.**

| Slope (°) | Slope pattern | Number of cultural heritages | Proportion (%) |
|---|---|---|---|
| 0–0.5 | Plain | 888 | 50.60 |
| 0.5–2 | Micro slope | 279 | 15.90 |
| 2–5 | Gentle slope | 256 | 14.59 |
| 5–15 | Slope | 318 | 18.12 |
| 15–25 | Steep slope | 14 | 0.80 |

**Table 7. Number and proportion of the Grand Canal (East Zhejiang section)-Maritime Silk Road cultural heritage with different slope aspects.**

| Slope aspect | Light conditions | Number of cultural heritages | Proportion (%) |
|---|---|---|---|
| −1 | Best | 77 | 4.39 |
| 90°–270° | Good | 845 | 48.15 |
| 45°–90° | Poor | 269 | 15.33 |
| 270°–315° | Poor | 130 | 7.41 |
| 0°–45° | Worst | 197 | 11.23 |
| 315°–360° | Worst | 237 | 13.50 |

(3) River and water system

Rivers are crucial water sources for human civilization. Ningbo has the Yongjiang River, one of the eight major water systems in Zhejiang Province, as well as other rivers such as the Yuyao River and Fenghua River. The Yuyao River and the Fenghua River converge at Three-River Estuary to form the Yongjiang River, which flows to the northeast and empties into the East China Sea at Zhaobao Mountain in Zhenhai. The Buffer tool in ArcGIS 10.8 software was used to establish different widths of river buffer zones by selecting rivers and streams within Ningbo as the water system data source (Fig 8d). Subsequently, cultural heritage points within these buffer zones were extracted (Table 8). The results show that within the 1-km river buffer zone, there are 922 cultural heritage points, accounting for 52.54% of the total; within the 2-km river buffer zone, there are 1,294 cultural heritage points, accounting for 73.73% of the total; within the 3-km river buffer zone, there are 1,524 cultural heritage points, accounting for 86.84% of the total; within the 5-km river buffer zone, there are 1,717 cultural heritage points, accounting for 97.83% of the total.

The results indicate that the cultural heritage of the Grand Canal (East Zhejiang section)-Maritime Silk Road is strongly hydrophilic in nature, showing a distribution along both sides of the river. During historical period, the proximity to the river ensured a sufficient water supply and convenient transportation, which promoted the formation and development of early settlements and cities [50].

**Human factors.**

(1) Political factor

The governance level of the national government reflects the economic status within a certain period and is related to the people's living conditions, thus affecting the quantity, type and characteristics of cultural heritage within that period. Table 9 presents a chronological analysis of the impact of political events on cultural heritage. It can be observed that during periods of political stability and social peace, such as the Tang, Ming, and Qing Dynasties, production developed rapidly and people lived and worked in peace and contentment, which led to a

**Table 8. Number and proportion of the Grand Canal (East Zhejiang section)-Maritime Silk Road cultural heritage with different widths of river buffer zones.**

| Buffer zone width (km) | Number of cultural heritages | Proportion (%) |
|---|---|---|
| 1 | 922 | 52.54 |
| 2 | 1294 | 73.73 |
| 3 | 1524 | 86.84 |
| 5 | 1717 | 97.83 |

**Table 9. Political events and impacts in different periods.**

| Periods | Political events and impacts |
|---|---|
| Qin Dynasty | China initiated the implementation of a unified and centralized system, yet the country's political and economic center of gravity remained in the north. Ningbo's remote location, inadequate governance, and sluggish economic development resulted in the preservation of a limited number of cultural heritages. |
| Tang dynasty | In 821, the Tang Dynasty established a prefectural seat in the Three-River Estuary, thereby laying the foundation for Ningbo's first true city. This development contributed to a favorable social and economic environment, which in turn facilitated the gradual accumulation of cultural heritage. |
| Song Dynasty | As the capital of the Southern Song Dynasty relocated to Lin'an (present-day Hangzhou, Zhejiang Province), China's political and economic center of gravity shifted southward, and Ningbo rapidly developed into an important commercial port and foreign trade center. This period saw a rapid increase in the number of cultural heritages, their coverage, and their types, which became increasingly diverse. |
| Ming Dynasty | At the beginning of the Ming Dynasty, the department of Shi Bo Si was established in Ningbo, which designated Ningbo as the sole port for tribute trade with Japan. This period saw remarkable developments in numerous domains, including politics, economics, and culture. Moreover, the number of cultural heritages increased rapidly. |
| Qing dynasty | As one of the five ports of commerce established following the Opium War, Ningbo absorbed a considerable amount of European and American capital, contributing to the diversification of its economy. During this period, the growth of cultural heritage reached its peak. |

large proportion of cultural heritage in these periods. Conversely, during periods of regime change, division, and constant wars and turmoil, such as the end of the Eastern Han Dynasty, the Wei, Jin and North and South Dynasties, the Five Dynasties and Ten Kingdoms, and the end of the Ming Dynasty and the beginning of the Qing Dynasty, production and living order were disrupted, thus fewer cultural heritages remained. In conclusion, the development of the cultural heritage of the Grand Canal (East Zhejiang section)-Maritime Silk Road was significantly influenced by national political factors. Periods of stability and prosperity fostered the creation and preservation of cultural heritage, while times of conflict and instability led to their decline.

(2) Population factor

In ancient societies, the level of science and technology was not sufficiently developed, resulting in low production efficiency. Consequently, population size often determined the mode of social production, as well as social organization and social structure [36]. The ruling classes of traditional Chinese societies regarded population size as one of the main symbols of national strength and social governance. More importantly, population is one of the main bases for the government to collect taxes and levy corvée [17]. As a product of people's production

and lifestyle in the historical period, the spatial and temporal distribution of cultural heritage is inevitably affected by the changes in population distribution and quantity. According to *Ningbo Population History* [51], the general trend of change in Ningbo's historical population can be described as follows: Ningbo's population grew slowly from the Qin Dynasty to the middle of the Tang Dynasty, reached its first peak in the Two Song Dynasties, declined considerably in the Yuan and Ming Dynasties, reached its second peak in the middle of the Qing Dynasty, and showed a curvilinear development in the Republic of China. This is broadly consistent with the changing characteristics of the number of cultural heritages in Ningbo. Furthermore, to more accurately reflect the quantitative relationship between the two variables, the population quantity in the *Ningbo Population History* was used to conduct a fitting analysis with the cultural heritage (Fig 9). This analysis revealed a significant correlation between the two variables, with $R^2 = 0.7786$, indicating that the change in population quantity could explain 77.86% of the change in the cultural heritage. Overall, the evolution pattern of Ningbo's population is largely consistent with the evolution trend of the spatial and temporal distribution of the cultural heritage of the Grand Canal (East Zhejiang section)-Maritime Silk Road. This suggests that population evolution has a significant impact on the spatial and temporal distribution of cultural heritage.

(3) Cultural factor

Cultural heritage is the material carrier of historical culture, and historical culture is the inner soul of cultural heritage. As a node city of China's Grand Canal and Maritime Silk Road, Ningbo has been influenced by the intersection of the Grand Canal culture and Maritime Silk Road culture, resulting in the formation of the regional culture of Ningbo, which is represented by Hemudu Culture, Ancient Yue Culture, Buddhist Culture, Zhedong Academic

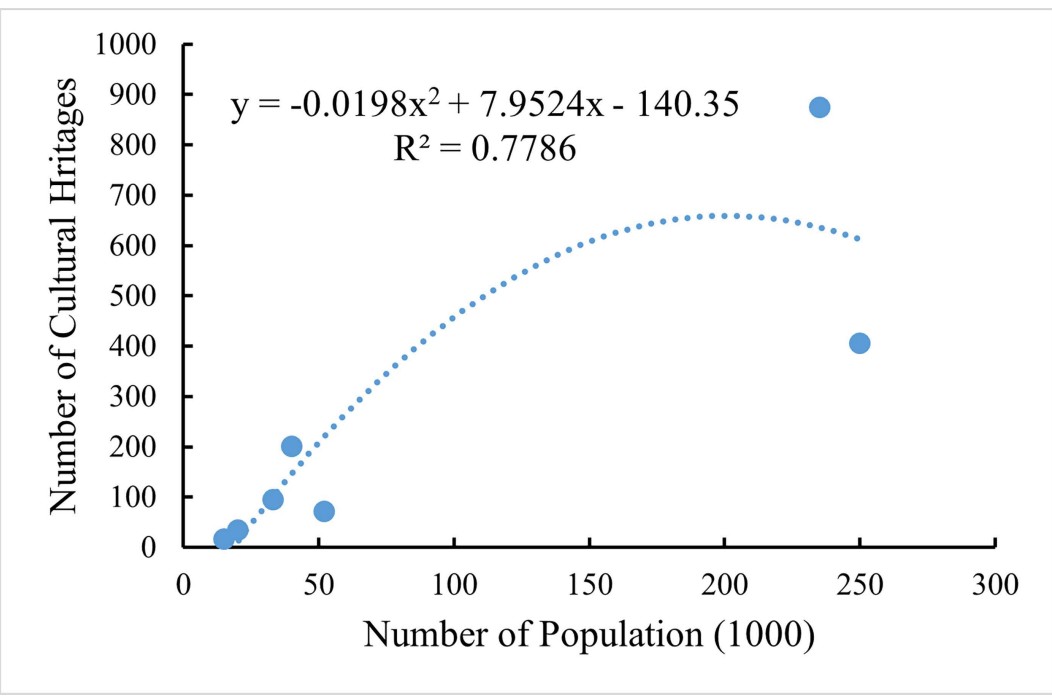

**Fig 9. Fitting curve between the number of population and the number of the Grand Canal (East Zhejiang section)-Maritime Silk Road cultural heritages in different periods.**

Culture, Book Collection Culture, Mazu Culture, Coastal Defense Culture, and Merchant Gang Culture over the course of history (Table 10). In the context of the humanistic background of the Grand Canal culture and Maritime Silk Road culture, the diverse regional cultures of Ningbo have contributed to the formation of a multitude of distinctive Grand Canal (East Zhejiang section)-Maritime Silk Road cultural heritage.

## Discussion

### Uniqueness of this study

Cultural heritage is a collective term for a multitude of tangible and intangible elements that reflect the past state of human existence, creativity, and the relationship between humans and the environment. These elements, which include monuments and artifacts, collectively form a testament to human civilization. This study analyzes the spatial and temporal distribution characteristics of cultural heritage in Ningbo, a city located at the intersection of the Grand Canal (East Zhejiang section) and the Maritime Silk Road, and explores the factors affecting this distribution.

The Grand Canal and the Maritime Silk Road, two of the most influential cultural routes in human history, have played a key role in the preservation of tangible cultural heritage,

**Table 10. Regional cultural impacts and representative cultural heritages.**

| Cultures | Period of emergence | Impacts | Representative cultural heritages |
|---|---|---|---|
| Hemudu Culture | Neolithic period | Hemudu Culture marks the cultural origins of the Yangtze River Basin, shaping an oceanic space that fostered a diverse Chinese civilization and laid the foundation for the Maritime Silk Road. | Marine culture sites, such as Jingtoushan Site. |
| Ancient Yue Culture | Warring States period | Ancient Yue Culture, a continuation of Hemudu Culture, is a maritime culture with an outward and dispersive nature, profoundly influencing the Maritime Silk Road. | Urban sites and tomb complexes, such as Gouzhang Former City Site. |
| Buddhist Culture | Late Eastern Han Dynasty | Buddhist Culture originated in India and spread to northern China and East Asia via the Grand Canal and the Maritime Silk Road. | Temple-type ancient buildings, such as Tiantong Temple, Ashoka Temple, and Baoguo Temple. |
| Zhedong Academic Culture | Northern Song Dynasty | The Zhedong school, led by Wang Yangming and Huang Zongxi, promoted pragmatism and the importance of industry and commerce, sparking early democratic ideas. These ideas spread via the Grand Canal and Maritime Silk Road, profoundly influencing regional culture and Ningbo's merchant prosperity. | Ancient buildings, such as Baiyun Village and Wang Yangming's Former Residence. |
| Book Collection Culture | Northern Song Dynasty | Ningbo's Book Collection Culture, shaped by private collections passed down through generations, reflects the city's long-standing pursuit of knowledge. | Book-collecting pavilions, such as Tianyi Pavilion, Wanjuan Tower, and Wugui Hall. |
| Mazu Culture | Late Northern Song Dynasty | Mazu, a goddess of the sea and key figure in China's maritime culture, is a central folk belief tied to the nation's peaceful diplomacy, maritime transport, and trade for over a thousand years. | Tianhou palace buildings, such as Qing'an Guild Hall. |
| Coastal Defense Culture | Ming Dynasty | Ningbo's Coastal Defense Culture, shaped by its resistance to British, French, and Japanese forces, is a vital part of Maritime Silk Road history. | Coastal defense sites, such as Zhenhaikou Coast Defense Site. |
| Merchant Gang Culture | Late Ming Dynasty | The Grand Canal and Maritime Silk Road fostered the rise of Ningbo Merchants, shaping their mercantilism, pioneering spirit, and global outlook. | Guild hall buildings, such as Qianye Guild Hall and Anlan Guild Hall. |

representing China's contribution to global civilization. Previous studies have primarily focused on intangible cultural heritage along the Grand Canal. For instance, Li [4] selected folk songs from the intangible cultural heritage of the Grand Canal and discovered that the spatial distribution of folk songs exhibited a pattern of "distribution along the river with two cores and two belts"; Wang et al. [52] confirmed that traditional skills in the Grand Canal Cultural Belt exhibits a spatial and temporal differentiation from multi-centered dispersal to centralized distribution at the two ends of the canal. However, there has been limited research on the spatial and temporal distribution of tangible cultural heritage, with only a few small data samples. Huang et al. [53] analyzed 104 cultural heritages in six periods along the Suzhou section of the Grand Canal and concluded that the canal's characteristics significantly shaped the distribution of these heritages. Studies on the cultural heritage along the Maritime Silk Road are even rarer. Niu et al. [54,55] focused on the marine cultural heritage of 11 coastal provinces and cities, exploring the composition of resources, spatial and temporal patterns, and influencing factors of the marine cultural heritage.

Globally, numerous cultural routes like France's Canal du Midi, Canada's Canal du Rideau, Oman's Frankincense Route, Egypt's Nile River, along with the Ancient Tea Horse Road and the Ancient Shu Road in China have also been studied, though primarily from archaeology [56,57], history [58], and tourism [59,60] perspectives. This study, however, focuses on the material heritage influenced by both the Grand Canal and the Maritime Silk Road, examining how their evolution and interplay have shaped the spatial and temporal distribution of cultural heritage in Ningbo.

The Grand Canal and the Maritime Silk Road, both crucial trade and cultural exchange routes in ancient China and globally, share a close interrelationship. This relationship is not only reflected in the geographical connection, but also in the complementarity and synergy between the two in promoting cultural and economic exchanges, which played an important role in the evolution and development of cultural heritage.

From a geographical standpoint, the Grand Canal and the Maritime Silk Road formed a comprehensive transportation network that connected inland and coastal areas. The Grand Canal facilitated the movement of goods and customs within China, while the Maritime Silk Road enabled international trade. This dual network not only encouraged economic and cultural exchanges but also shaped the diversity of Ningbo's cultural heritage, evident in its artistic and architectural styles.

From a cultural standpoint, the Grand Canal and the Maritime Silk Road served as conduits for the exchange between Chinese and global civilization. For instance, Indian Buddhism reached Ningbo via the Maritime Silk Road during the late Eastern Han Dynasty, fostering the development of historical Buddhist remains such as Tiantong Temple and Ashoka Temple [61]. Similarly, the spread of Mazu Culture, brought to Ningbo by the Fujian merchants via the Maritime Silk Road, extended throughout Southeast Asia and northern China. The Qing'an Guild Hall (A place for worshipping the sea goddess Mazu) in Ningbo, a symbol of Mazu Culture, reflects the convergence of canal and sea cultures and highlights the far-reaching influence of these routes on Ningbo's cultural heritage.

## Change and contemporary value of the cultural heritage of the grand canal and the maritime silk road

The culture of the Grand Canal and the Maritime Silk Road is a cross-regional and cross-generational phenomenon, integrating various cultures of the regions and countries along the route and leaving a substantial cultural legacy. The cultural heritage of the Grand Canal (East Zhejiang section)-Maritime Silk Road has evolved over time and space, with the help of the Grand Canal and the Maritime Silk Road, Chinese culture and Chinese civilization

have moved from the inland to the coasts and from the coasts to the sea. According to the composition structure of cultural heritage: there are a large number of ancient buildings and modern important historical sites and representative buildings, which have good cultural tourism functions and high potential for development and utilization, and are the key development objects of the cultural heritage of the Grand Canal (East Zhejiang section)-Maritime Silk Road. Ancient sites, ancient tombs and grotto temples and stone carvings as a whole are long-standing, rare and have deep cultural connotations. To ensure their development, it is crucial to prioritize their preservation.

The Grand Canal and the Maritime Silk Road are not only historically interconnected but have also maintained significant relevance in modern society. In ancient times, cultural heritage witnessed the history of the Grand Canal's access to rivers and the sea, and preserved the lifestyles, cultural beliefs, and ideologies of human beings in different historical periods. The Grand Canal and the Maritime Silk Road contributed to the development of cultural heritage, which intertwined with each other in the long course of history, and together constituted the open and tolerant cultural pattern of Chinese civilization with pluralism and coexistence. In modern times, the cultural heritage of the Grand Canal and the Maritime Silk Road has become a cultural bridge, connecting the ancient and the modern, and facilitating communication between the inside and the outside. The use of modern technology enables the presentation of cultural heritage in a more vivid and intuitive manner, thereby enhancing the understanding and appreciation of its historical and cultural values among a broader audience. Concurrently, the growth of the tourism industry has led to the development of special cultural tourism products that attract tourists to experience the charm of the Grand Canal and the Maritime Silk Road. This, in turn, has prompted the emergence of new forms of study tours, intangible cultural heritage experiences and digital tours, which have further facilitated the protection and inheritance of cultural heritage. Furthermore, the Belt and Road Initiative has facilitated international cooperation and exchange with regard to the cultural heritage of the Grand Canal and the Maritime Silk Road. By strengthening collaboration with international organizations, we can collectively excavate and protect cultural heritage, promote cultural exchanges and mutual understanding among countries and regions along the routes, and contribute to the formation of a community of human destiny.

## Future research directions

The data sources employed in the study are crucial in determining the conclusions drawn from the study. The cultural heritage items discussed in this study are drawn from the list published by the Cultural Heritage Administration. However, some of them are not closely related to the Grand Canal and the Maritime Silk Road due to their distance from these routes. For example, some modern red cultural heritage sites in the Siming Mountain Range are more affected by the wars and the topography than they are by the Grand Canal and the Maritime Silk Road. Such factors may influence the precision of the results to some extent. Additionally, the categorization of cultural heritage associated with the Grand Canal and the Maritime Silk Road can be further refined to explore common features of certain types of cultural heritage and improve study accuracy.

The accelerated construction of infrastructures such as the Internet, high-speed railways, and highways has facilitated the increased connectivity between cities. Many Chinese cities, including Suzhou, Yangzhou, Zhenjiang, and Lianyungang, have been influenced by the Grand Canal and the Maritime Silk Road throughout their historical development. Future research should examine the spatial and temporal distribution of cultural heritage in these cities, to identify similarities and differences in the influencing factors among different node cities, and strengthen the exchange and collaboration of cultural heritage development among these cities.

## Conclusions

This study analyzed the cultural heritage data of the Grand Canal (East Zhejiang Section)-Maritime Silk Road, examining the spatial and temporal distribution of different types of cultural heritage and the factors affecting them during historical period. Methods such as NNI, KDE, SDE, and buffer analysis were employed. The main conclusions are as follows:

(1) The cultural heritage spans all periods and includes various types of heritage. Ancient buildings and modern important historical sites and representative buildings are the most numerous, totaling 1,447 and accounting for 82.45%. This is followed by ancient sites, ancient tombs and grotto temples and stone carvings, with 143, 84 and 74 respectively. Others is the least, with only 7. In terms of periods, the lowest numbers were observed for prehistory to pre-Qin period (40) and Qin to Tang period (51). In contrast, Song to Yuan period exhibited a significant increasing trend (110), while Ming to Qing period and modern period exhibited the highest numbers (1,554), collectively accounting for 88.55% of the total. The overall trend of cultural heritage before modern period is upward, with a peak in Ming to Qing period.

(2) The cultural heritage shows a pattern of large agglomeration and small dispersion, with the Three-River Estuary as the core high-density area. Different types of heritage display distinct distribution patterns. Ancient buildings, sites, tombs, and grotto temples and stone carvings show an aggregation-random trend, while modern important historical sites and representative buildings exhibit aggregation, and others are randomly distributed. Ancient buildings and modern important sites and representative buildings' distribution mirrors the overall trend. Ancient sites cluster around the Yuyao Jiangbei junction, southern Haishu, and Dongqian Lake. Ancient tombs are more dispersed, with a core in northern Haishu. Grotto temples and stone carvings are concentrated in Haishu, the Three-River Estuary, and the Yinzhou Beilun junction. Overall, the heritage concentration is strongest near the Three-River Estuary, decreasing with distance.

(3) The distribution of cultural heritage varies by historical period. The closer the period, the more concentrated the heritage in the Three-River Estuary, particularly since the Ming to Qing periods. Pre-Qin heritage is mainly Neolithic settlements, while Qin-Tang heritage centers on Yue kiln sites. Song-Yuan heritage diversifies, mostly in north-central Ningbo. The Ming to Qing and modern periods are dominated by ancient buildings and important historical sites and representative buildings, all concentrated around the Three-River Estuary. The SDE azimuths range from 138° to 148.5°, indicating a northwest-southeast distribution of heritage. Over time, the center of gravity shifted from the south-southeast to west-north, converging toward the Three-River Estuary, with influence from the Grand Canal and Maritime Silk Road.

(4) The spatial-temporal distribution of the Grand Canal (East Zhejiang section)-Maritime Silk Road heritage is shaped by natural factors (topography, elevation, slope, slope aspect, rivers) and human factors (politics, population, culture). Heritage tends to cluster in low-elevation, gentle slope areas with strong hydrophilicity. However, the influence of slope aspect is negligible. While natural factors play a supporting role, human factors drive the evolution of heritage distribution.

## Supporting information

**S1 File. Cultural heritage dataset.**
(XLSX)

## Acknowledgments

First of all, we are grateful to Ningbo University for its support in the construction of the geography discipline, which ensured that our research could be carried out successfully.

Secondly, we would like to thank Qianyu Zha and Ke Jin for their great help in writing and revising my article, which laid the foundation for the successful completion and quality improvement of my article.

Finally, we would like to express my sincere thanks to the reviewers and editor who reviewed this paper and provided valuable comments.

## Author contributions

**Conceptualization:** Jie Li, Yuhe Gao, Chao Gao.

**Data curation:** Jie Li, Xinlian Yang, Yuhe Gao.

**Formal analysis:** Jie Li, Yuhe Gao.

**Funding acquisition:** Yuhe Gao, Chao Gao.

**Investigation:** Jie Li, Xinlian Yang.

**Methodology:** Jie Li, Chao Gao.

**Project administration:** Yuhe Gao.

**Resources:** Jie Li, Xinlian Yang.

**Software:** Jie Li, Xinlian Yang.

**Supervision:** Xinlian Yang, Yuhe Gao, Chao Gao.

**Validation:** Jie Li, Xinlian Yang, Chao Gao.

**Visualization:** Jie Li.

**Writing – original draft:** Jie Li.

**Writing – review & editing:** Yuhe Gao, Chao Gao.

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
