## [Decision Letter · Decision Letter 0]

4 Oct 2024

PONE-D-24-31693Spatial and temporal distribution characteristics and influencing factors of cultural heritage: a case of the Grand Canal (East Zhejiang section)-Maritime Silk RoadPLOS ONE

Dear Dr. Gao,

Thank you for submitting your manuscript to PLOS ONE. After careful consideration, we feel that it has merit but does not fully meet PLOS ONE’s publication criteria as it currently stands. Therefore, we invite you to submit a revised version of the manuscript that addresses the points raised during the review process.

The manuscript is innovative and valuable, focusing on the spatial-temporal distribution of cultural heritage related to the Grand Canal and Maritime Silk Road. Minor revisions are needed to improve data clarity, figure quality, structure, and language precision. Addressing these points will enhance the work's clarity and impact.

We look forward to receiving your revised manuscript.

Kind regards,

Elena Marrocchino

Academic Editor

PLOS ONE

“This work was funded by the Key Research and Development Program of Ningbo, Grant No. 2023Z137.”

“This work was funded by the Key Research and Development Program of Ningbo, Grant No. 2023Z137.”

“This work was funded by the Key Research and Development Program of Ningbo, Grant No. 2023Z137.”

5. We note that Figures 1, 2, 4, 5, 6, 7, 8 and 9 in your submission contain [map/satellite] images which may be copyrighted. All PLOS content is published under the Creative Commons Attribution License (CC BY 4.0), which means that the manuscript, images, and Supporting Information files will be freely available online, and any third party is permitted to access, download, copy, distribute, and use these materials in any way, even commercially, with proper attribution. For these reasons, we cannot publish previously copyrighted maps or satellite images created using proprietary data, such as Google software (Google Maps, Street View, and Earth). For more information, see our copyright guidelines: http://journals.plos.org/plosone/s/licenses-and-copyright.

1. You may seek permission from the original copyright holder of Figures 1, 2, 4, 5, 6, 7, 8 and 9 to publish the content specifically under the CC BY 4.0 license. 

Additional Editor Comments:

Based on the comments of all three reviewers, the manuscript presents a valuable and innovative analysis of the spatial and temporal distribution of cultural heritage, particularly in relation to the Grand Canal (East Zhejiang section) and the Maritime Silk Road. The study is of practical significance and is subject to a few key revisions. Minor adjustments are needed to improve the clarity of data sources, the formatting of formulae, the quality of figures, and the precision of language. In addition, suggestions include restructuring certain sections, refining the literature review, and enhancing the discussion of influencing factors with further references. Overall, the paper shows strong potential and would benefit from these refinements to increase clarity and impact.

Reviewers' comments:

Reviewer's Responses to Questions

**Comments to the Author**

1. Is the manuscript technically sound, and do the data support the conclusions?

Reviewer #1: Yes

Reviewer #2: Yes

Reviewer #3: Yes

2. Has the statistical analysis been performed appropriately and rigorously?

Reviewer #1: Yes

Reviewer #2: Yes

Reviewer #3: I Don't Know

3. Have the authors made all data underlying the findings in their manuscript fully available?

Reviewer #1: Yes

Reviewer #2: No

Reviewer #3: No

4. Is the manuscript presented in an intelligible fashion and written in standard English?

Reviewer #1: Yes

Reviewer #2: No

Reviewer #3: Yes

5. Review Comments to the Author

Reviewer #1: This paper focuses on the cultural heritage under the joint influence of the Grand Canal (East Zhejiang section) and the Maritime Silk Road, and explores its spatial and temporal evolution patterns and related influencing factors. This work shows a certain logic, substantive coherence, and has important practical significance. I will recommend it to be published at Plos ONE. However, the manuscript still has little issues that should be addressed through appropriate modifications. Thus, minor revision is needed.

1. L128: Please specify which part of the river system data was used.

2. L134: Please check the format of the formulas for each method in Research method and some of the formulas in its notes.

3. L235: The figure uses symbols of different shapes to mark out the different regions, which is not intuitive enough. It is recommended to mark them with boxes of different colors.

4. L235: Please keep the kernel density values in the figure to two decimals only and do the same for the subsequent kernel density figures.

5. L480: The third column of table 10 has too much to say about impacts, and it is suggested that cuts be made to streamline the sentences.

6. L628: Conclusions (3) and (4) have too many words and it is suggested to simplify them.

Reviewer #2: This manuscript analyzes the spatial-temporal distribution characteristics and influencing factors of cultural heritage, using the Grand Canal (East Zhejiang section) in China as a case study. The findings have positive implications for the high-quality and sustainable development of the Grand Canal and the Maritime Silk Road. Specific suggestions are as follows: (1) The description of the study area in the manuscript is unclear. For example, in the title, 'a case of the Grand Canal (East Zhejiang section) - Maritime Silk Road' is vague; (2) The quality of all figures is too low. It is recommended to optimize and enhance the figures, add latitude and longitude grids, and include other necessary elements; (3) Please add a description of the reliability of the underlying cultural heritage distribution data; (4) It is suggested to refine the language and expressions throughout the manuscript, as many parts contain obscure language and errors.

Reviewer #3: The article analyzes the characteristics of spatial and temporal distribution of cultural heritage from the perspective of geography, which is innovative and the results of the study are of practical significance. The regulations are clear and the logic is rigorous, but the manuscript still has the following problems, and it is suggested that it be revised and re-reviewed:

1. it is suggested to modify the overall structure by dividing the first introductory part into two parts, namely, research background and literature review, and combining the second part and the third part into a complete results part.

2. suggested deletions to the abstract

3. the second half of the research uniqueness section is too cumbersome and unfocused

4. the selection and categorization of influencing factors lacks some literature support, and it is suggested to add relevant influencing factors literature as a prelude.

6. PLOS authors have the option to publish the peer review history of their article (what does this mean? ). If published, this will include your full peer review and any attached files.

**Do you want your identity to be public for this peer review?** For information about this choice, including consent withdrawal, please see our Privacy Policy .

Reviewer #1: No

Reviewer #2: No

Reviewer #3: No

---

## [Author Response · Author response to Decision Letter 1]

12 Nov 2024

Reviewer #1: This paper focuses on the cultural heritage under the joint influence of the Grand Canal (East Zhejiang section) and the Maritime Silk Road, and explores its spatial and temporal evolution patterns and related influencing factors. This work shows a certain logic, substantive coherence, and has important practical significance. I will recommend it to be published at Plos ONE. However, the manuscript still has little issues that should be addressed through appropriate modifications. Thus, minor revision is needed.

1. L128: Please specify which part of the river system data was used.

Response: Thanks for the heads up, this omission has been added.

The Digital Elevation Model (DEM) data and the river data were obtained from the Geospatial Data Cloud (http://www.gscloud.cn/) and the Open Street Map (https://www.openstreetmap.org/), respectively. In addition, for the river data, irrelevant factors such as drains and sewers were eliminated and naturally occurring rivers were selected.

2. L134: Please check the format of the formulas for each method in Research method and some of the formulas in its notes.

Response: Thanks for the heads up. The formatting of the equation section in the research method has all been rechecked. Both the numbered equations and inline equations have been edited by applying the Mathtype, which is consistent with the formatting sample provided by Plos ONE. Therefore, there is no need to revise the section.

3. L235: The figure uses symbols of different shapes to mark out the different regions, which is not intuitive enough. It is recommended to mark them with boxes of different colors.

Response: Thanks to your suggestion, we have redrawn Figure 4.

Fig 4. Overall kernel density of the Grand Canal (East Zhejiang section)-Maritime Silk Road cultural heritage. (The blue box indicates Three-River Estuary, the green box indicates Yuyao city center, and the red box indicates Xiangshan Shipu fishing port.)

4. L235: Please keep the kernel density values in the figure to two decimals only and do the same for the subsequent kernel density figures.

Response: Thank you for your suggestions. Because some densities are so small that retaining two decimals is not appropriate, we have retained two significant figures for all density values.

Fig 4. Overall kernel density of the Grand Canal (East Zhejiang section)-Maritime Silk Road cultural heritage. (The blue box indicates Three-River Estuary, the green box indicates Yuyao city center, and the red box indicates Xiangshan Shipu fishing port.)

Fig 5. Kernel density of different types of the Grand Canal (East Zhejiang section)-Maritime Silk Road cultural heritage.

Fig 6. Kernel density of the Grand Canal (East Zhejiang section)-Maritime Silk Road cultural heritage in different periods.

5. L480: The third column of table 10 has too much to say about impacts, and it is suggested that cuts be made to streamline the sentences.

Response: This part has been redacted.

Table 10. Regional cultural impacts and representative cultural heritages.

Cultures Period of emergence Impacts Representative cultural heritages

Hemudu Culture Neolithic period Hemudu Culture marks the cultural origins of the Yangtze River Basin, shaping an oceanic space that fostered a diverse Chinese civilization and laid the foundation for the Maritime Silk Road. Marine culture sites, such as Jingtoushan Site.

Ancient Yue Culture Warring States period Ancient Yue Culture, a continuation of Hemudu Culture, is a maritime culture with an outward and dispersive nature, profoundly influencing the Maritime Silk Road. Urban sites and tomb complexes, such as Gouzhang Former City Site.

Buddhist Culture Late Eastern Han Dynasty Buddhist Culture originated in India and spread to northern China and East Asia via the Grand Canal and the Maritime Silk Road. Temple-type ancient buildings, such as Tiantong Temple, Ashoka Temple, and Baoguo Temple.

Zhedong Academic Culture Northern Song Dynasty The Zhedong school, led by Wang Yangming and Huang Zongxi, promoted pragmatism and the importance of industry and commerce, sparking early democratic ideas. These ideas spread via the Grand Canal and Maritime Silk Road, profoundly influencing regional culture and Ningbo’s merchant prosperity. Ancient buildings, such as Baiyun Village and Wang Yangming’s Former Residence.

Book Collection Culture Northern Song Dynasty Ningbo’s Book Collection Culture, shaped by private collections passed down through generations, reflects the city’s long-standing pursuit of knowledge. Book-collecting pavilions, such as Tianyi Pavilion, Wanjuan Tower, and Wugui Hall.

Mazu Culture Late Northern Song Dynasty Mazu, a goddess of the sea and key figure in China’s maritime culture, is a central folk belief tied to the nation’s peaceful diplomacy, maritime transport, and trade for over a thousand years. Tianhou palace buildings, such as Qing’an Guild Hall.

Coastal Defense Culture Ming Dynasty Ningbo’s Coastal Defense Culture, shaped by its resistance to British, French, and Japanese forces, is a vital part of Maritime Silk Road history. Coastal defense sites, such as Zhenhaikou Coast Defense Site.

Merchant Gang Culture Late Ming Dynasty The Grand Canal and Maritime Silk Road fostered the rise of Ningbo Merchants, shaping their mercantilism, pioneering spirit, and global outlook. Guild hall buildings, such as Qianye Guild Hall and Anlan Guild Hall.

6. L628: Conclusions (3) and (4) have too many words and it is suggested to simplify them.

Response: Thanks for your suggestions, we have simplified conclusions (3) and (4), and additionally (2).

(2) The cultural heritage shows a pattern of large agglomeration and small dispersion, with the Three-River Estuary as the core high-density area. Different types of heritage display distinct distribution patterns. Ancient buildings, sites, tombs, and grotto temples and stone carvings show an aggregation-random trend, while modern important historical sites and representative buildings exhibit aggregation, and others are randomly distributed. Ancient buildings and modern important sites and representative buildings’ distribution mirrors the overall trend. Ancient sites cluster around the Yuyao Jiangbei junction, southern Haishu, and Dongqian Lake. Ancient tombs are more dispersed, with a core in northern Haishu. Grotto temples and stone carvings are concentrated in Haishu, the Three-River Estuary, and the Yinzhou Beilun junction. Overall, the heritage concentration is strongest near the Three-River Estuary, decreasing with distance.

(3) The distribution of cultural heritage varies by historical period. The closer the period, the more concentrated the heritage in the Three-River Estuary, particularly since the Ming to Qing periods. Pre-Qin heritage is mainly Neolithic settlements, while Qin-Tang heritage centers on Yue kiln sites. Song-Yuan heritage diversifies, mostly in north-central Ningbo. The Ming to Qing and modern periods are dominated by ancient buildings and important historical sites and representative buildings, all concentrated around the Three-River Estuary. The SDE azimuths range from 138° to 148.5°, indicating a northwest-southeast distribution of heritage. Over time, the center of gravity shifted from the south-southeast to west-north, converging toward the Three-River Estuary, with influence from the Grand Canal and Maritime Silk Road.

(4) The spatial-temporal distribution of the Grand Canal (East Zhejiang section)-Maritime Silk Road heritage is shaped by natural factors (topography, elevation, slope, slope aspect, rivers) and human factors (politics, population, culture). Heritage tends to cluster in low-elevation, gentle slope areas with strong hydrophilicity. However, the influence of slope aspect is negligible. While natural factors play a supporting role, human factors drive the evolution of heritage distribution.

Reviewer #2: This manuscript analyzes the spatial-temporal distribution characteristics and influencing factors of cultural heritage, using the Grand Canal (East Zhejiang section) in China as a case study. The findings have positive implications for the high-quality and sustainable development of the Grand Canal and the Maritime Silk Road. Specific suggestions are as follows:

(1) The description of the study area in the manuscript is unclear. For example, in the title, ‘a case of the Grand Canal (East Zhejiang section) - Maritime Silk Road’ is vague;

Response: We have modified it in the Research area section to make it seem clearer.

Located on the Ningshao Plain in the eastern part of Zhejiang Province (Fig. 1), Ningbo, as the southernmost city of the Grand Canal and a key starting port of the Maritime Silk Road, holds a unique historical position. At the opening ceremony of the Belt and Road Forum for International Cooperation, General Secretary Xi Jinping stated that the ancient ports in Ningbo and other places are “living fossils” that record the history of the ancient Silk Road. Ningbo has a long history and is the birthplace of the Hemudu Culture, which represents over 7,000 years of civilization [30]. Due to its unique historical background and geographic location, Ningbo has developed a series of cultures with regional characteristics, mainly the Grand Canal culture and the Maritime Silk Road culture. As a carrier of cultural dissemination and historical memory, the cultural heritage in Ningbo bears the imprints of different eras, making it quintessential area for researching the cultural heritage of the Grand Canal and the Maritime Silk Road. As a carrier of cultural transmission and historical memory, cultural heritage carries the imprints of different eras. Therefore, Ningbo is selected in this study as a case to explore the cultural heritage of the Grand Canal (East Zhejiang section)-Maritime Silk Road.

(2) The quality of all figures is too low. It is recommended to optimize and enhance the figures, add latitude and longitude grids, and include other necessary elements;

Response: Thank you for your suggestion, we have optimised and redrawn all the figures, while including the necessary elements. These figures will be uploaded separately in the submission system.

(3) Please add a description of the reliability of the underlying cultural heritage distribution data;

Response: Thanks for the heads up, we have added the description.

Data attributes of cultural heritage include name, era, type, latitude and longitude coordinates, address, protection level, and other relevant information of the cultural heritage. The geographic coordinates of cultural heritage sites were primarily collected using Global Positioning System (GPS) during field trips. Some hard-to-reach cultural heritage sites had their coordinates obtained by applying Baidu map coordinate picking. Additionally, Google Earth was utilized to cross-check these coordinates. Excel and ArcGIS10.8 software were used to establish the database of the cultural heritage of the Grand Canal (East Zhejiang section)-Maritime Silk Road and create the spatial distribution map of the cultural heritage (Fig. 2).

(4) It is suggested to refine the language and expressions throughout the manuscript, as many parts contain obscure language and errors.

Response: Thank you for your suggestions, we have rechecked and revised the writing style throughout.

In Abstract:

The Grand Canal and the Maritime Silk Road in China are globally significant cultural routes, which have contributed a wealth of cultural heritage through their historical development.

Focusing on the Ningbo area, this study analyzes the spatial and temporal distribution of 1,755 cultural heritage sites over five historical periods and explores the influencing factors through spatial and statistical analysis.

The results show that: (1) Ancient buildings, along with modern important historical sites and representative buildings, are the most numerous.

In Introduction:

The distribution of cultural heritage worldwide reflects the complex pathways of human social development, geographic changes, and historical evolution, showcasing diverse temporal and spatial characteristics.

This study will help clarify the development of these two cultural routes and foster cultural preservation.

In Materials and methods:

Located on the Ningshao Plain in the eastern part of Zhejiang Province (Fig. 1), Ningbo, as the southernmost city of the Grand Canal and a key starting port of the Maritime Silk Road, holds a unique historical position.

The geographic coordinates of cultural heritage sites were primarily collected using Global Positioning System (GPS) during field trips.

Some cultural heritage items are listed at both the national and provincial levels, and some single items include multiple geographically distant sites. To more accurately study the spatial distribution characteristics of the cultural heritage, this study treats duplicated items as a single item, and single items containing multiple cultural heritages as multiple items.

The SDE captures the central tendency, dispersion, and directional patterns of spatial elements, enabling the analysis of cultural heritage distribution over different periods [34].

In Result:

The elevation statistics of the cultural heritage sites reveal a distinct distribution pattern (Table 5).

In Discussion:

The Grand Canal and the Maritime Silk Road, both crucial trade and cultural exchange routes in ancient China and globally, share a close interrelationship.

The Grand Canal and the Maritime Silk Road are not only historically interconnected but have also maintained significant relevance in modern society.

Reviewer #3: The article analyzes the characteristics of spatial and temporal distribution of cultural heritage from the perspective of geography, which is innovative and the results of the study are of practical significance. The regulations are clear and the logic is rigorous, but the manuscript still has the following problems, and it is suggested that it be revised and re-reviewed:

1. it is suggested to modify the overall structure by dividing the first introductory part into two parts, namely, research background and literature review, and combining the second part and the third part into a complete results part.

Response: Thank you for your suggestions; we have adjusted the structure of the article. Because the structure of the article is not easy to show, please see the word document called ‘Revised Manuscript with Track Changes’ for details.

2. suggested deletions to the abstract

Response: Thank you for your suggestions, we have condensed the abstract.

The Grand Canal and the Maritime Silk Road in China are globally significant cultural routes, which have contributed a wealth of cultural heritage through their historical development. The study on the cultural heritage of the Grand Canal (East Zhejiang section)-Maritime Silk Road is of great significance for constructing the Grand Canal Cultural Belt and advancing the Belt and Road Initiative. Focusing on the Ningbo area, this study analyzes the spatial and temporal distribution of 1,755 cultural heritage sites over five historical periods and explores the influencing factors through spatial and statistical analysis. The results show that: (1) Ancient buildings, along with modern important historical sites and representative buildings, are the most numerous. The total number of cultural heritages shows an upward trend before the modern period, peaking in the Ming to Qing period. (2) The cultural heritage exhibits an overall aggregated spatial distribution, with varying patterns across different types. The Three-River Estuary is the high-density core area, with the number and density of cultural heritage decreasing as its distance increases. (3) Distribution characteristics of cultural heritage vary across different periods. More recent cultural heritage is increasingly concentrated around the Three-River Estuary. Over time, the center of gravity of cultural heritage has shifted sequentially to the south, southeast, west, and north. (4) The cultural heritage tends to be distributed in plain with low altitude and small slope, and shows strong hydrophilicity. However, it is essentially influenced by human factors.

3. the second half of the researc

---

## [Decision Letter · Decision Letter 1]

17 Jan 2025

Spatial and temporal distribution characteristics and influencing factors of cultural heritage: a case of the Grand Canal (East Zhejiang section)-Martime Silk Road

PONE-D-24-31693R1

Dear Dr. Gao,

We’re pleased to inform you that your manuscript has been judged scientifically suitable for publication and will be formally accepted for publication once it meets all outstanding technical requirements.

Kind regards,

Peter F. Biehl, PhD

Academic Editor

PLOS ONE

Additional Editor Comments (optional):

Reviewers' comments:

Reviewer's Responses to Questions

**Comments to the Author**

1. If the authors have adequately addressed your comments raised in a previous round of review and you feel that this manuscript is now acceptable for publication, you may indicate that here to bypass the “Comments to the Author” section, enter your conflict of interest statement in the “Confidential to Editor” section, and submit your "Accept" recommendation.

Reviewer #2: All comments have been addressed

Reviewer #3: (No Response)

2. Is the manuscript technically sound, and do the data support the conclusions?

Reviewer #2: Yes

Reviewer #3: (No Response)

3. Has the statistical analysis been performed appropriately and rigorously?

Reviewer #2: Yes

Reviewer #3: (No Response)

4. Have the authors made all data underlying the findings in their manuscript fully available?

Reviewer #2: Yes

Reviewer #3: (No Response)

5. Is the manuscript presented in an intelligible fashion and written in standard English?

Reviewer #2: Yes

Reviewer #3: (No Response)

6. Review Comments to the Author

Reviewer #2: The author has arranged suggestions to revise the manuscript, and I believe that the current manuscript has met the acceptance criteria. I suggest accepting it.

Reviewer #3: (No Response)

7. PLOS authors have the option to publish the peer review history of their article (what does this mean? ). If published, this will include your full peer review and any attached files.

**Do you want your identity to be public for this peer review?** For information about this choice, including consent withdrawal, please see our Privacy Policy .

Reviewer #2: No

Reviewer #3: No

---

## [Editor Report · Acceptance letter]

PONE-D-24-31693R1

PLOS ONE

Dear Dr. Gao,

I'm pleased to inform you that your manuscript has been deemed suitable for publication in PLOS ONE. Congratulations! Your manuscript is now being handed over to our production team.

Kind regards,

on behalf of

Dr. Peter F. Biehl

Academic Editor

PLOS ONE